# Functional links between sensory representations, choice activity, and sensorimotor associations in parietal cortex

Ting-Yu Chang[1†], Raymond Doudlah[1†], Byounghoon Kim[1], Adhira Sunkara[2], Lowell W Thompson[1], Meghan E Lowe[1], Ari Rosenberg[1]*

[1]Department of Neuroscience, School of Medicine and Public Health, University of Wisconsin–Madison, Madison, United States; [2]WiSys Technology Foundation, Madison, United States

**Abstract** Three-dimensional (3D) representations of the environment are often critical for selecting actions that achieve desired goals. The success of these goal-directed actions relies on 3D sensorimotor transformations that are experience-dependent. Here we investigated the relationships between the robustness of 3D visual representations, choice-related activity, and motor-related activity in parietal cortex. Macaque monkeys performed an eight-alternative 3D orientation discrimination task and a visually guided saccade task while we recorded from the caudal intraparietal area using laminar probes. We found that neurons with more robust 3D visual representations preferentially carried choice-related activity. Following the onset of choice-related activity, the robustness of the 3D representations further increased for those neurons. We additionally found that 3D orientation and saccade direction preferences aligned, particularly for neurons with choice-related activity, reflecting an experience-dependent sensorimotor association. These findings reveal previously unrecognized links between the fidelity of ecologically relevant object representations, choice-related activity, and motor-related activity.

**\*For correspondence:**
ari.rosenberg@wisc.edu

[†]These authors contributed equally to this work

**Competing interests:** The authors declare that no competing interests exist.

## Introduction

Interactions with the environment require the mapping of sensory information to motor responses. The parietal cortex is an important site of sensorimotor transformations (*Rushworth et al., 1997*; *Buneo et al., 2002*; *Brovelli et al., 2004*; *Buneo and Andersen, 2006*). Parietal lesions can result in deficits associated with the impaired use of sensory information to create and execute motor plans, as opposed to deficits in sensory processing or action (*Pause and Freund, 1989*; *Pause et al., 1989*). However, they can also produce 3D visual processing deficits (*Holmes, 1918*; *Holmes and Horrax, 1919*). Although sensorimotor transformations are widely studied using two-dimensional (2D) stimulus paradigms, mappings between 3D object information and motor responses are essential in the natural world. Thus, parietal cortex may have a fundamental role in creating robust 3D representations and mapping them to specific actions.

Within parietal cortex, a major site of 3D visual processing is the caudal intraparietal (CIP) area. Neurons in CIP are tuned for 3D orientation (*Taira et al., 2000*; *Rosenberg et al., 2013*) signaled by multiple cues (*Tsutsui et al., 2001*; *Tsutsui et al., 2002*; *Rosenberg and Angelaki, 2014b*) and perform multisensory processing required for gravity-centered vision (*Rosenberg and Angelaki, 2014a*). In addition to high-level sensory representations, CIP activity correlates with the short-term memory and perceptual matching of 3D features (*Tsutsui et al., 2003*) and choices made during a

binary orientation discrimination task (*Elmore et al., 2019*). Furthermore, inactivating CIP impairs 3D feature discrimination (*Tsutsui et al., 2001*; *Van Dromme et al., 2016*).

The anatomical projections and effective connectivity of CIP suggest that it may contribute to goal-directed sensorimotor transformations (*Nakamura et al., 2001*; *Premereur et al., 2015*; *Van Dromme et al., 2016*; *Lanzilotto et al., 2019*). In particular, CIP projects to areas involved in motor planning and execution through eye and hand movements, including the lateral intraparietal (LIP) area (*Andersen et al., 1992*; *Bennur and Gold, 2011*; *Shushruth et al., 2018*), anterior intraparietal (AIP) area (*Murata et al., 2000*; *Baumann et al., 2009*; *Pani et al., 2014*), and V6A (*Fattori et al., 2010*; *Fattori et al., 2012*; *Breveglieri et al., 2016*). However, it is unknown if CIP neurons carry motor-related signals, and if so, whether they form sensorimotor associations.

Here we investigated the relationships between sensory representations, choice-related activity, and motor-related activity in CIP. Neuronal activity was recorded while macaque monkeys performed an eight-alternative forced choice (8AFC) tilt discrimination task with planar surfaces at different slants and distances (*Chang et al., 2020*). The neurons differed in the extent to which their orientation tuning depended on distance. Choice-related activity was preferentially carried by neurons whose 3D orientation selectivity was more tolerant to distance and associated with further increases in tolerance. Neuronal choice tuning was parametric and the choice and surface tilt preferences generally aligned. A comparison of results from monkeys trained to report opposite stimulus features (i.e. the near versus far side of the plane; *Elmore et al., 2019*) indicated that the alignment was experience dependent. Motor-related activity was also assessed for the first time in CIP using a visually guided saccade task (*Munoz and Wurtz, 1995*; *Hanes and Schall, 1996*). Many neurons had saccade direction tuning and the saccade direction and surface tilt preferences generally aligned. This sensorimotor association was strongest for neurons with choice-related activity.

These results reveal systematic relationships between sensory representations, choice-related activity, and motor-related activity in CIP. Neurons with more robust 3D representations (i.e. those whose 3D orientation selectivity was more tolerant to distance) were most strongly coupled to behavior. Beyond traditional associations between neuronal choice-related activity and decision processes, our findings identify links to the fidelity of high-level sensory representations and the strength of sensorimotor associations. The coupling of sensory and motor functions at the level of single CIP neurons may facilitate sensorimotor transformations over short timescales that enable successful interactions with the 3D world.

## Results

We tested three hypotheses about the relationships between sensory representations, choice-related activity, and motor-related activity. First, that neurons with more robust 3D representations would preferentially carry choice-related activity. Second, that choice-related activity would be associated with further increases in the robustness of 3D selectivity, not the attenuation of information not directly relevant to the task at hand (i.e. nuisance variables). Third, that sensorimotor associations would be experience dependent and stronger for neurons with choice-related activity.

### Quantifying behavioral sensitivity to 3D surface tilt

Testing our hypotheses required a protocol that could be used to simultaneously measure the 3D visual sensitivity of the monkeys as well as quantify the 3D selectivity of individual neurons. To this end, we developed an 8AFC tilt discrimination task in which the side of a viewed plane that was nearest was reported through a saccade to one of eight choice targets (*Figure 1A*). We then trained two monkeys to perform this task under different viewing conditions that determined the task difficulty (*Chang et al., 2020*). Specifically, by presenting planes at multiple combinations of tilt, slant, and distance we could assess how 3D visual sensitivity and neuronal responses depended on 3D surface pose (i.e. orientation and position).

The 3D orientation of a planar surface is described by tilt and slant (*Figure 1B*). Tilt ($0° \leq T < 360°$) specifies the direction that the plane is oriented in depth. For example, $T = 0°$ indicates right-near and $T = 90°$ indicates top-near. Planes were presented at eight tilts (0° to 315°, 45° steps), corresponding to the eight choice options. Slant ($0° \leq S \leq 90°$) specifies the amount of depth variation. Larger slants indicate greater depth variation. Because a frontoparallel plane ($S = 0°$) has no depth variation, tilt is undefined at that orientation. As such, frontoparallel planes were ambiguous stimuli

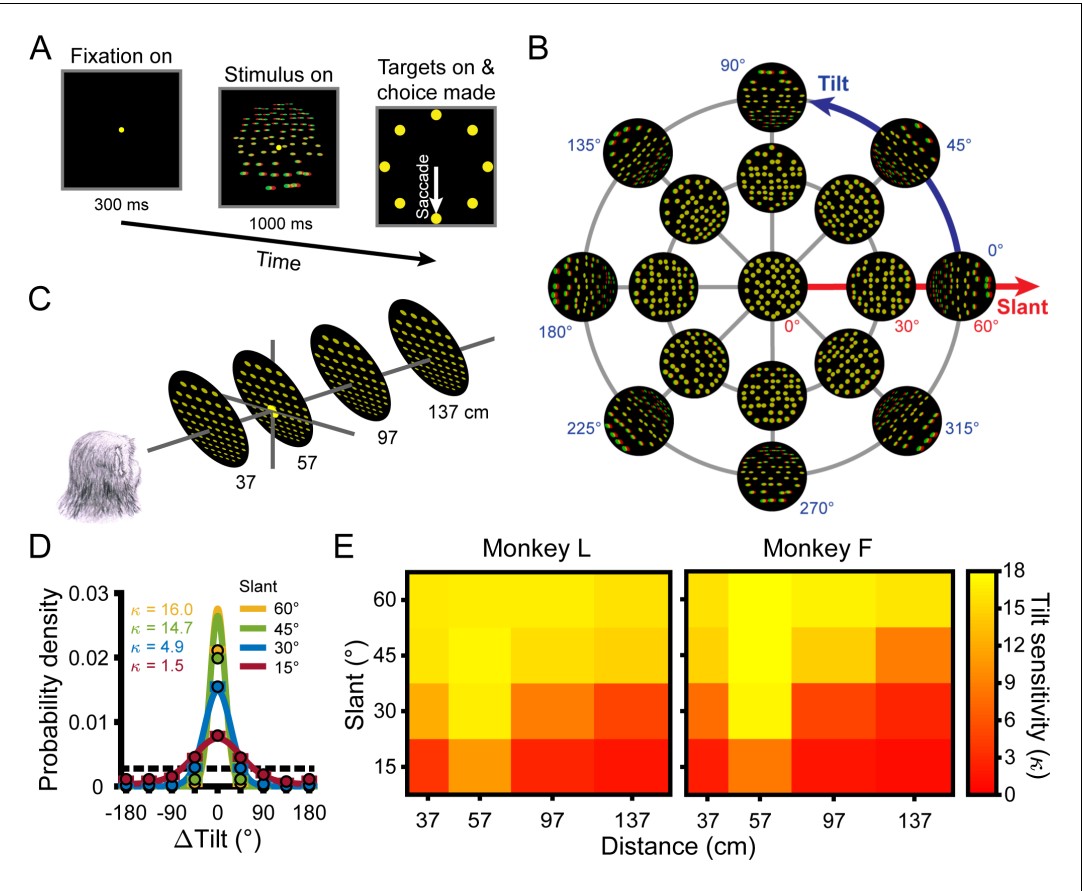

**Figure 1.** Task, stimuli, and performance. (**A**) Tilt discrimination task. After fixating a target at the center of the screen for 300 ms, a plane was shown for 1 s while fixation was maintained. The target and plane then disappeared and eight choice targets appeared. The nearest side of the plane was then reported via a saccade to the corresponding target (e.g. the bottom target for a bottom-near plane). (**B**) Tilt and slant. Planes were rendered as random dot stereograms with perspective and stereoscopic cues. A subset of planes are shown here as red–green anaglyphs. For clarity, the number of dots is reduced and the dot size is increased compared to the actual stimuli. (**C**) Planes were presented at the center of the screen at four distances and always subtended 20° of visual angle. The fixation point (depicted by the larger yellow dot) was always located at 57 cm. (**D**) Error distributions of reported tilts at each slant (distance = 137 cm) calculated over all tilts for Monkey L. Data points show the mean probability of a given ΔTilt (correct choice: ΔTilt = 0°) and SEM (error bars are obscured by the data points) across sessions. Solid curves are von Mises probability density functions. Taller and narrower densities indicate greater sensitivity. Sensitivities (κ, the von Mises concentration parameter) are indicated in the inset. At higher sensitivities, deviations between data points at ΔTilt = 0° and the density functions reflect discrete versus continuous representations of the area between presented tilts (*Chang et al., 2020*). Chance performance is marked by the dashed horizontal line. (**E**) Heat maps show mean tilt sensitivity across sessions as a function of slant and distance for each monkey. Yellow hues indicate greater sensitivity.

The online version of this article includes the following figure supplement(s) for figure 1:

**Figure supplement 1.** Behavioral performance.

in the task and no correct response could be objectively defined. Planes were presented at five slants (0° to 60°, 15° steps). All combinations of these tilts and slants (N = 33 unique orientations) were presented at four distances: 37, 57, 97, and 137 cm (*Figure 1C*). Thus, we measured behavioral and neuronal responses to 132 surface poses.

Behavioral performance was quantified for each combination of tilt, slant, and distance using the distribution of reported tilt errors (ΔTilt = reported tilt – presented tilt) (*Chang et al., 2020*). The sensitivity to each stimulus condition was defined as the concentration parameter (κ) of the von Mises probability density function fit to the corresponding error distribution (*Equation 1*) and was calculated for each session (Monkey L: N = 26; Monkey F: N = 27). For both monkeys, we found that sensitivity significantly depended on slant (four-way ANOVA, both $p \leq 6.0 \times 10^{-11}$) and distance (both $p \leq 1.3 \times 10^{-34}$) but not tilt (linearized into cosine and sine components, all four $p \geq 0.21$; *Fisher, 1995*). We therefore summarized sensitivity as a function of slant and distance

calculated over all tilts (*Figure 1D*, *Figure 1—figure supplement 1*). Consistent with our previous findings in which performance in the 8AFC tilt discrimination task was extensively analyzed over a wider range of 3D poses and multiple visual cue conditions (*Chang et al., 2020*), sensitivity decreased with distance from fixation (57 cm) and increased with slant (*Figure 1E*).

## The representation of 3D surface pose by CIP neurons

We next needed to quantify the representation of 3D surface pose in CIP. Although previous studies examined the 3D orientation selectivity of CIP neurons (*Taira et al., 2000*; *Tsutsui et al., 2001*; *Tsutsui et al., 2002*; *Tsutsui et al., 2003*; *Rosenberg et al., 2013*; *Rosenberg and Angelaki, 2014a*; *Rosenberg and Angelaki, 2014b*; *Elmore et al., 2019*), it is currently unknown how that selectivity depends on distance. This information is important because it can distinguish 3D representations from sensitivity to lower-level features (e.g. local binocular disparity cues) that co-vary with changes in object pose (*Janssen et al., 2000*; *Nguyenkim and DeAngelis, 2003*; *Alizadeh et al., 2018*; *Elmore et al., 2019*). To disentangle these possibilities, we measured how the 3D orientation tuning depended on the surface's distance with the fixation distance held constant. With this protocol, robust 3D selectivity is indicated if the orientation selectivity (tuning curve shape) is tolerant to the distance (though the gain can change, reflecting distance tuning). In contrast, sensitivity to lower-level features is implied if the selectivity substantially changes with distance. To quantify the representation of 3D surface pose in CIP while the monkeys performed the behavioral task, we recorded from 437 neurons using laminar array probes (Monkey L: N = 218; Monkey F: N = 219; *Figure 2*).

Sensory and choice-related signals co-exist in CIP but follow distinct time courses (*Elmore et al., 2019*). In *Figure 3—figure supplement 1A*, we show response time courses for an example neuron and the population to preferred and non-preferred stimuli. The median visual response latency was 52 ms and the population responses to preferred and non-preferred stimuli became significantly different at 58 ms (ANOVA, p<0.05), indicating that selectivity for 3D surface pose started early in the response. The onset of choice-related activity in the population response followed at 202 ms (calculated in a subsequent section). To test our hypotheses regarding associations between 3D selectivity and choice-related activity, we operationally defined two time windows for performing analyses. The first window captured the sensory dominant (SD) activity and was defined from the median visual response latency to the onset of choice-related activity in the population response. We confirm that

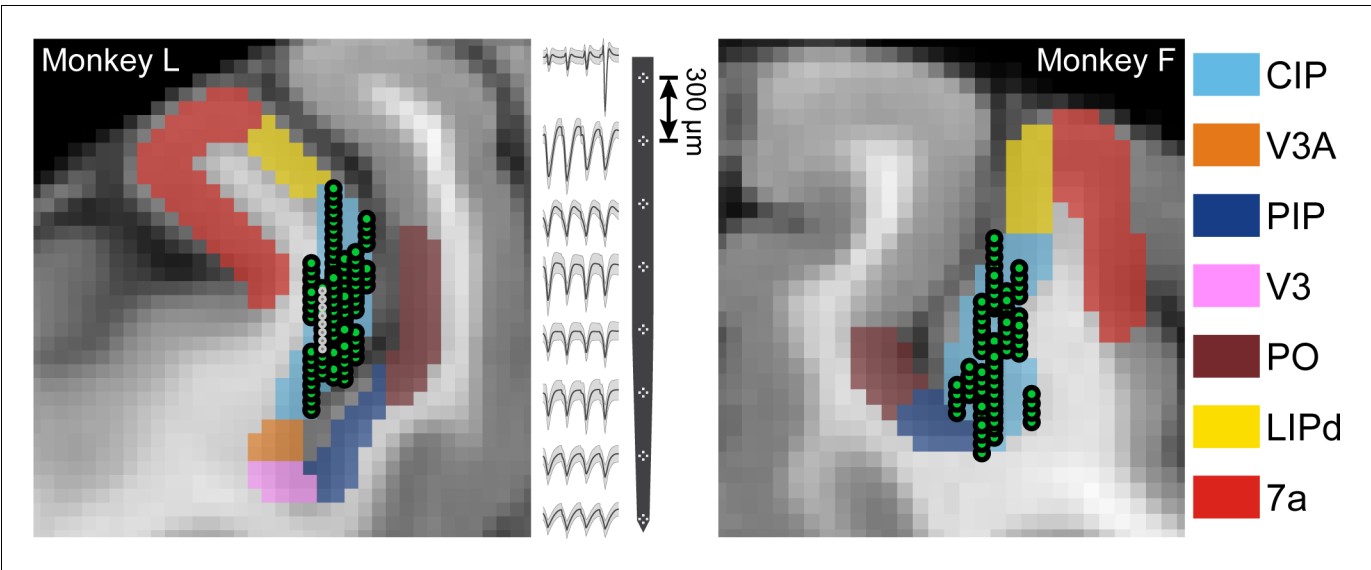

**Figure 2.** Neuronal recordings. Coronal MRI sections showing the estimated boundaries of CIP (light blue) and neighboring regions (see Materials and methods). All recording locations (green circles) are shown projected along the anterior-posterior (AP) axis onto a single section (Monkey L: AP = −7 mm; Monkey F: AP = −5.5 mm). Spike waveforms from an eight-tetrode recording (locations marked by smaller white circles in the MRI) are shown for Monkey L.

choice-related activity was not detectable within the SD window in a subsequent section, but nevertheless refer to this window as 'sensory dominant' to reflect the possibility that some choice-related activity (e.g. at the tail end of the window) could have gone undetected. The second window contained sensory plus choice (SPC) activity in the population response and was defined from the onset of choice-related activity to the end of the stimulus presentation. In this section, we quantify the representation of 3D surface pose during the SD window. In a subsequent section, we characterize how the representation differed in the SPC window. The 3D pose tuning curves of five neurons measured during the SD window are shown in *Figure 3*. These example neurons are used throughout the paper to portray the range of sensory, choice-related, and motor-related properties we found in CIP.

In an idealized 3D representation, the orientation tuning curve shape (e.g. preference and bandwidth, but not amplitude) is independent of distance, indicating that the 3D pose tuning curve is separable over orientation and distance. To quantify how well CIP activity conformed to this ideal, we fit each neuron's responses with a separable model (*Equation 8*). The model captured how well a neuron's joint tuning for orientation and distance was described by the multiplication of a 3D orientation tuning curve and a distance tuning curve (this model performed significantly better than an additively separable model for every neuron, see Materials and methods). A Tolerance index was then defined as the average correlation between the responses and fit at each distance. Values near one indicate that the shape of the orientation tuning curves was more tolerant to distance. Values near 0 indicate that the shape changed substantially with distance, as expected for neurons sensitive to lower-level visual features. Some neurons were highly tolerant, indicating robust 3D pose tuning (*Figure 3A*; Tolerance = 0.88). Others had modest tolerances, suggesting intermediate representations between lower-level features and 3D pose (*Figure 3B*; Tolerance = 0.53). Yet others had low

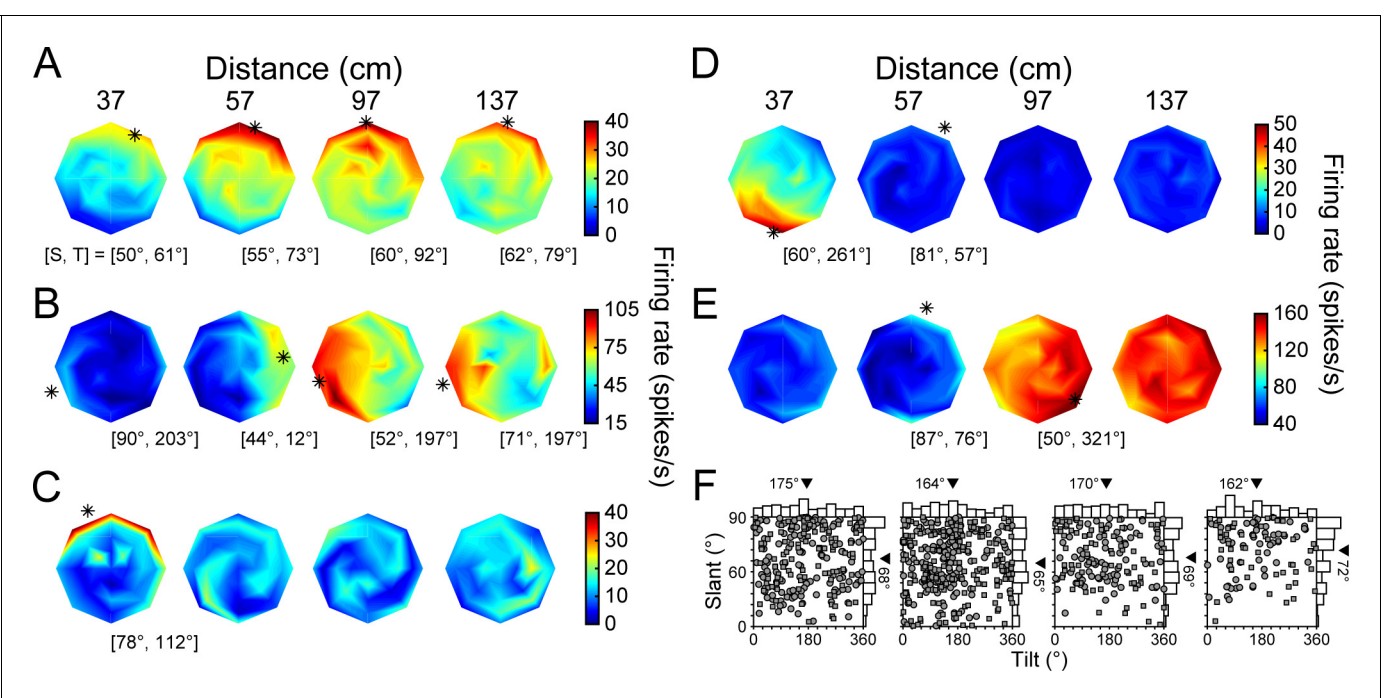

**Figure 3.** 3D orientation tuning at each distance before the onset of choice-related activity. (A-E) Five example neurons. Heat maps show firing rate as a function of tilt (T) and slant (S), plotted using the coordinates illustrated in *Figure 1B*. Red hues indicate higher firing rates. Asterisks mark preferred orientations from Bingham fits (tuned cases only) (*Rosenberg et al., 2013*). Some asterisks are off of the discs because the largest tested slant was 60°. (A) High tolerance neuron (Tolerance = 0.88). (B) Intermediate tolerance neuron (Tolerance = 0.53). (C) Low tolerance neuron (Tolerance = 0.34). (D) Low tolerance neuron (Tolerance = 0.24). (E) Distance selective neuron with little orientation-related modulation. (F) Distribution of T and S preferences at each distance plotted using an equal area projection (Monkey L: circles; Monkey F: squares). Only tuned neurons at each distance are included: 37 (N = 242), 57 (N = 320), 97 (N = 158), 137 (N = 106) cm. Marginal histograms show the T and S distributions. Triangles mark mean values.

The online version of this article includes the following figure supplement(s) for figure 3:

**Figure supplement 1.** Response time course and Tolerance distributions.

tolerances, consistent with lower-level feature selectivity (*Figure 3C,D*; Tolerance = 0.34 and 0.24, respectively). A minority had stronger distance than orientation selectivity (*Figure 3E*). Indeed, 62 neurons (14%) were distance (two-way ANOVA, p<0.05) but not orientation (p≥0.05) selective. Across the population, the mean Tolerance was 0.58 ± 0.15 standard deviation (Std Dev; *Figure 3— figure supplement 1B*).

The 3D orientation preferences of CIP neurons were previously found to be uniformly distributed when measured at the fixation distance (*Rosenberg et al., 2013*). To test if this finding held across surface distance (with constant fixation distance), preferences were estimated by fitting a Bingham function to each significant orientation tuning curve (ANOVA, p<0.05, Bonferroni-Holm corrected for each neuron). Consistent with the previous result, the distribution of tilt preferences was not significantly different from uniform at any distance ($\chi^2$; 37, 57, and 97 cm: p≥0.43; 137 cm: p=0.03, not significant after Bonferroni-Holm correction; *Figure 3F*). Likewise, the distribution of slant preferences at 57 cm (fixation distance) was not significantly different from uniform (p=0.13). However, the distributions of slant preferences were significantly different from uniform at 37, 97, and 137 cm (all p≤1.2x10$^{-3}$ and Bonferroni-Holm corrected). At each of these distances, the mean slant preference across the population was larger than at 57 cm. We therefore wanted to identify the neuronal factors underlying this dependency.

One possibility was that the slant preferences of individual neurons increased with distance from fixation. To test this, we performed pairwise comparisons of the slant preferences of individual neurons with significant orientation tuning at adjacent distances (within-neuron analysis). Indeed, the preferences tended to increase between 57 and 97 cm (mean ΔSlant = 5.3°; paired t-test, p=1.1×10$^{-3}$, N = 144 tuned at both distances) as well as between 97 and 137 cm (mean ΔSlant = 4.3°; p=9.7×10$^{-3}$, N = 87 tuned at both distances). These results indicate an inseparability in the 3D pose tuning such that the slant preferences tended to increase with distance behind fixation. The slant preferences of individual neurons were also larger at 37 than 57 cm, but the difference was not significant (mean ΔSlant = 2.1°; p=0.19, N = 209 tuned at both distances). This finding suggests a second (not mutually exclusive) possibility that neurons tuned for distances away from fixation tended to prefer larger slants. To test this, we compared the slant preferences of neurons tuned at 57 cm only (N = 75) to the slant preferences of the combined set of neurons tuned at 37, 97, or 137 only (N = 31). Consistent with this possibility, the mean slant preference of neurons tuned at a single distance away from fixation was 7.6° greater than for neurons tuned only at fixation (Wilcoxon rank sum test, p=0.02). Thus, the larger mean slant preferences at distances away from fixation reflected an inseparability in the 3D pose tuning of individual neurons as well as a tendency for neurons that represented distances away from fixation to prefer larger slants.

## Neuronal correlates of 3D tilt sensitivity

We next wanted to compare the behavioral and neuronal findings. Two prominent features of the behavioral data were increasing tilt sensitivity as a function of slant and an inverted U-shape relationship between tilt sensitivity and distance that peaked at the fixation distance (*Chang et al., 2020*; *Figure 1E*, *Figure 1—figure supplement 1*). These findings prompted us to test if behavioral tilt sensitivity was related to the ability to discriminate preferred from non-preferred tilts using individual neuron responses (during the SD window to minimize potential choice-related activity). For each neuron and slant–distance combination, we calculated a tilt discrimination index (TDI) that quantified the strength of response modulation across tilts relative to the overall response variability (*Prince et al., 2002*; *Elmore et al., 2019*; *Equation 3*). The behavioral tilt sensitivities (averaged over sessions) and TDI values (averaged over neurons) followed similar trends: both increased as a function of slant (*Figure 4A*) and had an inverted U-shape relationship with distance (*Figure 4B*). Indeed, across all sixteen slant–distance combinations, the tilt sensitivities and TDI values were highly correlated (Monkey L: r = 0.95, p=2.2×10$^{-308}$; Monkey F: r = 0.93, p=2.2×10$^{-308}$). Thus, the 3D tilt sensitivity of the monkeys was closely related to the tilt discriminability of CIP neurons over a wide range of viewing conditions.

We previously found that behavioral tilt sensitivity was consistent with a neuronal probabilistic population code model that integrated perspective and stereoscopic cues to 3D orientation (*Chang et al., 2020*). That model predicted a monotonic relationship between behavioral tilt sensitivity and the population response amplitude. To test this prediction, we compared the average tilt sensitivity at each slant–distance combination to the corresponding amplitude of the tilt tuning

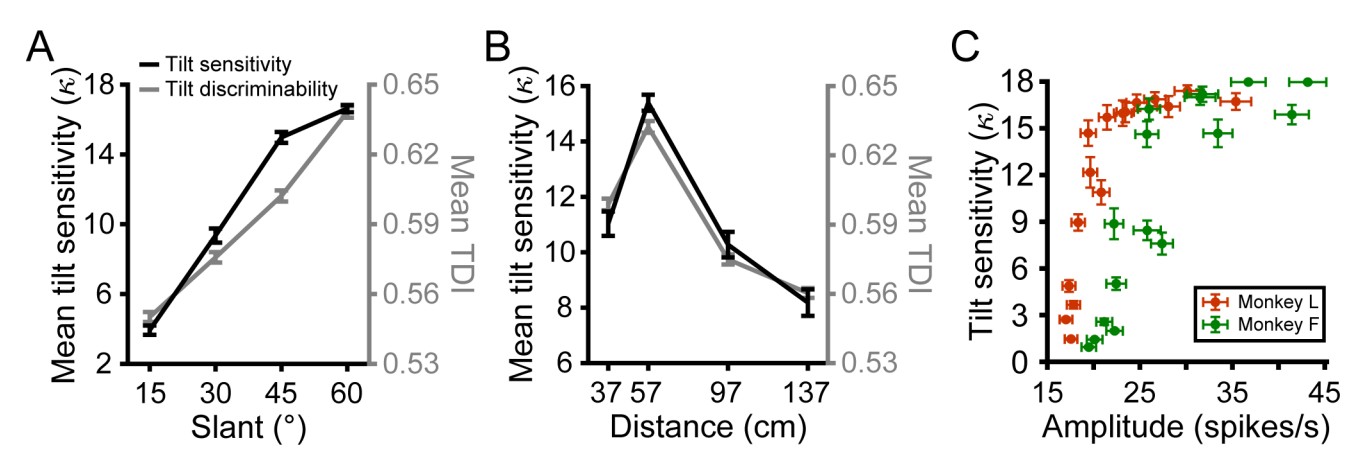

**Figure 4.** Neuronal correlates of 3D tilt sensitivity. (A) Mean behavioral tilt sensitivity ($\kappa$) and neuronal tilt discrimination index (TDI) values during the SD window increased with slant. Data points show mean and SEM across distances and monkeys or neurons. (B) Mean tilt sensitivity and TDI values during the SD window had an inverted U-shape relationship with distance that peaked at the fixation distance (57 cm). Data points show mean and SEM across slants and monkeys or neurons. (C) Tilt sensitivity versus response amplitude during the SD window at each slant–distance combination (Monkey L: orange; Monkey F: green). Data points show mean and SEM across sessions or neurons. Tilt sensitivity does not exceed $\kappa$ = 18, the upper limit that can be estimated with a 45° tilt sampling interval (*Chang et al., 2020*).

curves averaged over all neurons during the SD window (*Figure 4C*). Consistent with the prediction, tilt sensitivity increased as a function of response amplitude (Monkey L: Spearman r = 0.95, p=2.2×10$^{-308}$; Monkey F: r = 0.87, p=2.2×10$^{-308}$). This result parallels the strong correlation between tilt sensitivity and TDI value (which depends on response modulation and variability). The finding that the neuronal responses predicted behavioral sensitivity across performance levels ranging from near-chance to near-perfect suggests that CIP activity may constrain the precision of tilt perception.

## Choice-related activity was parametrically tuned and aligned with the tilt preferences

To test our hypotheses regarding the fidelity of 3D visual representations and choice-related activity, we needed to classify the neurons according to whether or not they carried choice-related activity. To ensure that we distinguished choice-related activity from tilt-selective responses, we performed the classification using responses to frontoparallel planes (S = 0°) only. Because tilt is undefined at that orientation, the stimuli were task-ambiguous. To increase the statistical power, we combined responses across all distances after separately z-scoring them at each distance. Responses were then grouped according to the choice made on each trial.

An advantage of the 8AFC task was that it could reveal parametric choice tuning, which is not possible with typical binary choice tasks. We therefore computed eight population-level time courses relative to the choice that elicited the maximum response for each neuron (*Figure 5A*). The responses first showed an initial response transient that was sensory dominant and therefore untuned since only frontoparallel plane responses (z-scored at each distance) were included. They then began to show parametric tuning with amplitudes that fell off symmetrically with greater deviation from the preferred choice (note the similarity of the ±45°, ±90°, and ±135° curves). The onset of choice-related activity (202 ms) was defined as the first time point that the time courses significantly diverged (ANOVA, p<0.05). As discussed above, this time point was used to operationally define two response epochs: a sensory dominant (SD) window and a sensory plus choice (SPC) window. All choice analyses were performed using data from the SPC window.

Across the population, 201 neurons (46%) had significant choice-related activity (ANOVA, p<0.05). Choice tuning curves with von Mises fits are shown for the example neurons in *Figure 5B–F* (blue curves). The high (*Figures 3A* and *5B*), intermediate (*Figures 3B* and *5C*), and first low (*Figures 3C* and *5D*) Tolerance neurons all had choice-related activity. However, the second low

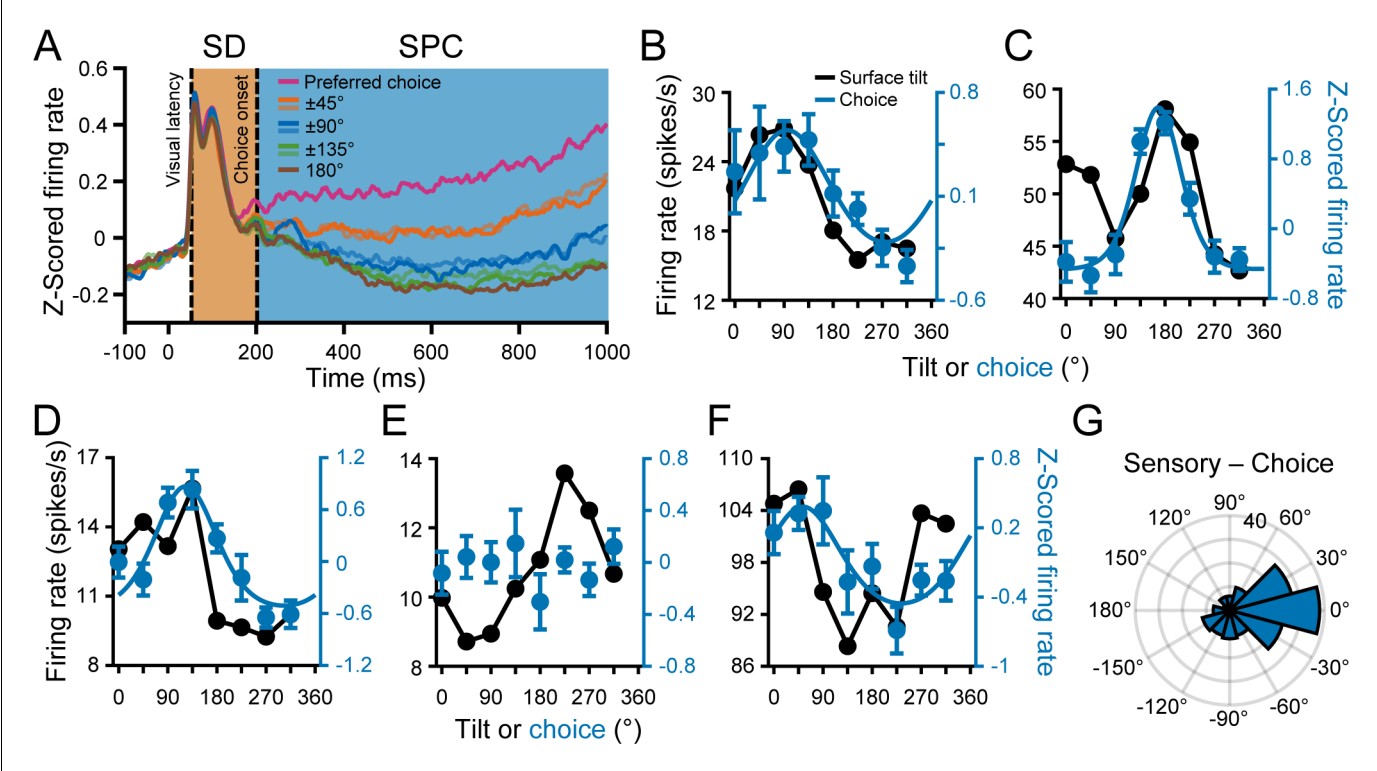

**Figure 5.** Choice tuning was parametric and aligned with the tilt preferences. (A) Time course of choice-related activity. Curves show z-scored responses averaged over neurons for each of the eight choices relative to each neuron's preferred choice. Stimulus onset = 0 ms. Vertical dashed lines mark the median visual response latency (52 ms) and onset of choice-related activity (202 ms). Shaded regions mark the SD (orange) and SPC (blue) windows. (B-F) Comparison of surface tilt and choice tuning for the example neurons (*Figure 3A–E*, same order). Black points are surface tilt responses marginalized over slant and distance (note that these responses were measured during the SD window, before the onset of choice-related activity). The tilt tuning curve in C has two peaks because the orientation preference was distance dependent. Blue points are z-scored choice responses and the curves are von Mises fits (tuned cases only). (G) Difference between the principal surface tilt (SD window only) and choice preferences (N = 166). The peak at 0° indicates that sensory and choice preferences generally aligned.

The online version of this article includes the following figure supplement(s) for figure 5:

**Figure supplement 1.** Validation of the choice-related activity results.

**Figure supplement 2.** Experience-dependent association of choice reports and orientation selectivity.

Tolerance neuron did not (*Figures 3D* and *5E*). The neuron with stronger distance than orientation tuning also had choice-related activity (*Figures 3E* and *5F*). Across the population, the choice tuning curves were well described by von Mises functions (mean r = 0.88 ± 0.10 Std Dev, N = 201). The mean concentration parameter was $\kappa$ = 3.52 ± 4.74 Std Dev (half-width at half-height, HWHH: 52° ± 21° Std Dev).

For comparison, surface tilt tuning curves measured during the SD window and marginalized over slant and distance are also shown in *Figure 5B–F* (black curves). The surface tilt and choice preferences of the example neurons aligned, even for the neuron with stronger distance than orientation tuning. To quantify this relationship across the population, we used the orientation preferences measured at each distance to compute a principal orientation preference for each neuron (SD window only; see Materials and methods and Figure 7D). We then compared the principal surface tilt and choice preferences from the von Mises fits (*Figure 5G*). The median circular difference between the preferences was −8.3° and not significantly different from 0° (circular median test, p=0.24; N = 166 neurons with orientation and choice tuning), indicating that the sensory and choice preferences generally aligned.

Because the SD and SPC windows were defined using a population-level measure, a potential concern was that they were not sensitive to individual neuron differences in the onset of choice-related activity, which could inadvertently introduce choice-related activity into the SD window. We

therefore tested each neuron for choice-related activity in the SD window (ANOVA, p<0.05). Consistent with the expected rate of false positives, the result of this test was significant for only 5% of neurons (22/437, none were example neurons) and not associated with the classification of having choice-related activity since only 13 had significant choice tuning in the SPC window. Moreover, excluding those 22 neurons did not significantly affect the mean Tolerances of neurons with or without choice-related activity in either time window (t-test, all four p≥0.64). These findings suggest that the SD and SPC windows defined reliable, functionally distinct response epochs at the individual neuron level. Another potential concern was that the SPC window was longer than the SD window. We therefore tested a short SPC window (850 ms – 1 s) that matched the duration of the SD window (150 ms) and found similar results (*Figure 5—figure supplement 1A-D*). Thus, the full SPC window was sufficient to characterize choice tuning and its relationship to surface tilt selectivity. To further validate the time-course of choice-related activity, we performed demixed principal component analysis and classified the behavioral choices using the choice projections (*Kobak et al., 2016*; *Figure 5—figure supplement 1E,F*). Following the cross-validation procedures of Kobak and colleagues, the classification accuracy did not exceed chance level until 232 ms, providing a second estimate of the onset of choice-related activity. Note that we used the more stringent estimate of 202 ms in our analyses. Together, these findings corroborated the distinction between the SD and SPC windows.

We wanted to further evaluate if the alignment of preferred surface tilts and choice report directions reflected a naturally occurring correspondence or a learned association. The current experiment was designed to distinguish between these possibilities through a comparison with results from another recent CIP study (*Elmore et al., 2019*). The two studies trained monkeys to report opposite features of viewed planar surfaces. For this study, we trained monkeys to report the near side of the plane. In contrast, the previous study trained monkeys to report the far side. This difference allowed us to test if the alignment of preferred surface tilts and choice report directions was experience dependent since opposite choice report directions were required for the same surface tilt. Importantly, we found that the alignment was reversed in the two data sets (*Figure 5—figure supplement 2*). For example, consider a typical neuron preferring a bottom-near plane (T = 270°). If the monkey was trained to report the near side, downward (upward) choice reports were associated with stronger (weaker) neuronal responses. Thus, a preference for bottom-near planes aligned with a preference for downward choice reports (which, based on the training, were used to indicate T = 270°). In contrast, if the monkey was trained to report the far side, upward (downward) choice reports were associated with stronger (weaker) responses. Thus, a preference for bottom-near planes aligned with a preference for upward choice reports (which, based on that training, were used to indicate T = 270°). Thus, how a neuron's preferred choice report direction aligned with its surface tilt preference depended on the task training. This comparison indicates that the association between 3D orientation selectivity and choice report directions did not simply reflect a natural correspondence but was instead flexible and experience dependent.

## Choice-related activity was associated with more robust 3D selectivity

The above analyses identified response epochs that were either sensory dominant (SD) or contained sensory plus choice (SPC) activity, and two neuronal subpopulations based on whether or not they had choice-related activity. These distinctions allowed us to next test our hypotheses regarding the fidelity of 3D visual representations and choice-related activity by comparing the 3D pose tuning curves across the time windows and subpopulations. To illustrate how the tuning curves changed across the time windows, the example neurons' 3D pose tuning curves measured during the SPC window are shown in *Figure 6A–E*. First, consider the three neurons with orientation and choice tuning. The selectivity of the neuron with high Tolerance in the SD window changed little and its Tolerance slightly increased: ΔTolerance = 0.08 (*Figures 3A* and *6A*). Strikingly, the orientation tuning of the neuron with intermediate Tolerance in the SD window aligned across all four distances in the SPC window to match the tuning at the preferred distance (*Figures 3B* and *6B*). Its Tolerance greatly increased: ΔTolerance = 0.35. For the neuron with low tolerance in the SD window, responses at non-preferred distances remained relatively weak in the SPC window (*Figures 3C* and *6C*). However, the orientation tuning was now significant at each distance and the preferences were similar, resulting in a large increase in Tolerance: ΔTolerance = 0.46. Next, consider the neuron with low Tolerance during the SD window and no choice-related activity (*Figures 3D* and *6D*). The selectivity of this

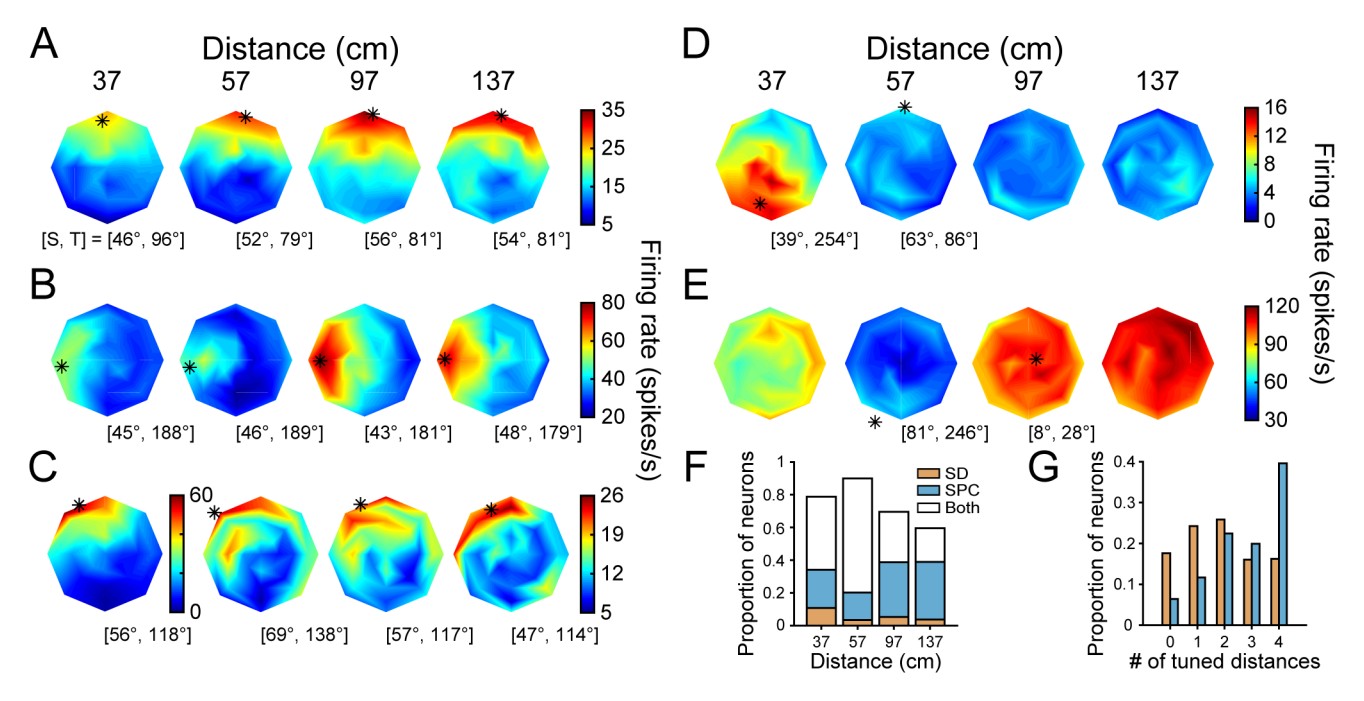

**Figure 6.** 3D orientation tuning at each distance after the onset of choice-related activity. (A-E) The example neurons (*Figures 3A–E* and *5B–F*, same order). (A) Tolerance increased from 0.88 (SD window) to 0.96 (SPC window). (B) Tolerance increased from 0.53 to 0.88. Note the changes in selectivity at 37 and 57 cm. (C) Tolerance increased from 0.34 to 0.80. The number of distances with significant orientation tuning increased from one to four. (D) Tolerance increased from 0.24 to 0.32. (E) Distance selective neuron. (F) Proportion of neurons with orientation tuning at each distance during the SD window only (orange), SPC window only (blue), or both windows (white). The proportion of tuned neurons decreased with distance from fixation (57 cm), consistent with the behavioral sensitivity (*Figures 1E* and *4B*). (G) Proportion of neurons with orientation tuning at each possible number of distances during the SD (orange) and SPC (blue) windows.

neuron changed relatively little between the time windows: ΔTolerance = 0.08. Lastly, the more distance selective neuron continued to respond most strongly to the furthest distances (*Figures 3E* and *6E*).

The changes in the example neurons' 3D pose tuning curves across the time windows were representative of the population. In particular, more neurons showed significant orientation tuning at each distance in the SPC window than the SD window (ANOVA, p<0.05; *Figure 6F*). Correspondingly, the neurons tended to have orientation tuning at more distances in the SPC window than the SD window (*Figure 6G*). There was also a reduction in the number of neurons which were distance but not orientation selective in the SPC window (two-way ANOVA; N = 20, 5%) compared to the SD window (N = 62, 14%). Consistent with a shift towards a higher-level 3D pose representation, the mean SPC window Tolerance (0.66 ± 0.16 Std Dev, N = 437) was greater than the mean SD window Tolerance (0.58 ± 0.15 Std Dev), and the difference was statistically significant (paired t-test, p=1.3×10$^{-18}$; *Figure 3—figure supplement 1C*).

Having identified two subpopulations of neurons based on whether or not they carried choice-related activity, we were able to further test our hypothesis that neurons with more robust 3D selectivity would preferentially carry choice-related activity. To do so, we used the Tolerance values from the SD window (i.e. before the onset of choice-related activity) to compare the 3D visual representations of neurons with versus without choice-related activity (*Figure 7A*). Consistent with our hypothesis, neurons with choice-related activity had a higher mean Tolerance (0.61) than those without choice-related activity (0.56), and the difference was statistically significant (ANOVA followed by Tukey's HSD test, p=0.02). Thus, neurons with choice-related activity tended to have more robust 3D selectivity than neurons without choice-related activity. Importantly, this difference in the 3D selectivity of the two subpopulations was present in the SD window, indicating that it could not be attributed to choice-related activity.

Neuroscience

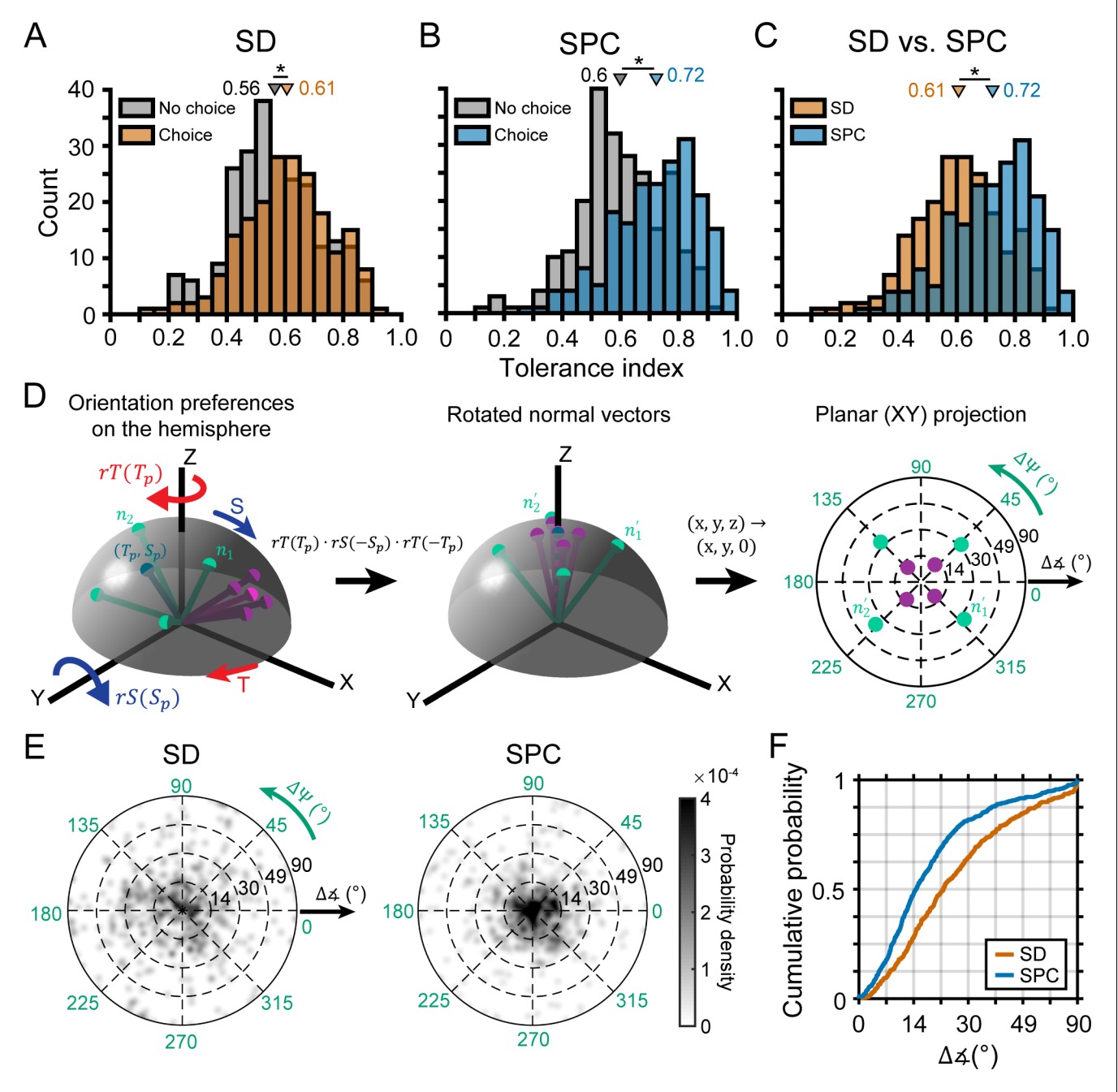

**Figure 7.** Choice-related activity was associated with more robust 3D selectivity. (A) Tolerance values in the SD window for neurons with (orange) and without (gray) choice-related activity. (B) Tolerance values in the SPC window for neurons with (blue) and without choice-related activity. (C) Tolerance values across the SD and SPC windows for neurons with choice-related activity. In A-C, triangles mark mean values. Asterisks indicate significant differences. (D) Calculating deviations between orientation preferences. Left: Hemispherical representation of surface orientation (*Rosenberg et al., 2013*). Slant and tilt correspond to the elevation and azimuth, respectively. Each ball and stick corresponds to the normal vector of a planar surface ($n_i$; *Equation 4*). Orientation preferences at four distances are shown for two hypothetical neurons with low (green) and high (purple) Tolerances (note the different spread in orientation preferences). Principal orientations (tilt = $T_p$, slant = $S_p$; see Materials and methods) are shown in different shades. Middle: Normal vectors rotated to align the principal orientation with the north pole ($n'_i$; *Equations 5-7*). Right: Equatorial projection of the rotated normal vectors, providing a standardized space that describes deviations in the orientation preferences from the principal orientation (direction: $\Delta\Psi$; angle: $\Delta\sphericalangle$). The origin corresponds to no difference and the outer ring to the maximum difference (90°). (E) Probability densities for the deviations in orientation preference during the SD and SPC windows (Gaussian smoothing kernel, $\sigma = 0.025$). (F) Cumulative density functions for the angular deviations (radial distances in E) for the SD and SPC windows. In C,E,F, only neurons with choice-related activity are included.

*Figure 7 continued on next page*

*Figure 7 continued*

The online version of this article includes the following figure supplement(s) for figure 7:

**Figure supplement 1.** Parametric analysis of 3D orientation tuning curve shape.

We next wanted to test our hypothesis that choice-related activity would be associated with further increases in the robustness of 3D selectivity (i.e. that 3D orientation selectivity would become more tolerant to distance). To start, we compared each neuron's 3D pose tuning in the SD and SPC windows by calculating the correlation over all 132 poses. The mean correlation was r = 0.49 ± 0.27 Std Dev (N = 437), indicating that the tuning curves were generally similar but not identical. Besides our hypothesis, an alternative explanation for this relatively modest average correlation could be the attenuation of selectivity for slant and distance (nuisance variables in the task), leaving only selectivity for the task-relevant variable (tilt). Contrary to this possibility, slant and distance tuning persisted in the SPC window for the example neurons (*Figure 6A–E*). To quantify this observation, we tested each neuron for slant and distance tuning (four-way ANOVA with slant, distance, and tilt linearized into cosine and sine components, p<0.05) in each time window. The number of neurons with slant tuning was 334 (76%) in the SD window and 370 (85%) in the SPC window. The number of neurons with distance tuning was 414 (95%) in the SD window and 412 (94%) in the SPC window. Because the prevalence of slant and distance tuning did not decrease during the SPC window (indeed, orientation selectivity increased; *Figure 6F,G*), these results are inconsistent with choice-related activity being associated with the attenuation of information not directly relevant to the task at hand.

Having ruled out that choice-related activity was associated with the attenuation of selectivity for slant and distance, we tested our hypothesis that it would be associated with an increase in the robustness of 3D selectivity. Specifically, that 3D orientation selectivity would be more tolerant to distance in the SPC than the SD window. First, we compared the mean Tolerance values between neurons with (0.72) and without (0.60) choice-related activity in the SPC window and found that the difference was statistically significant (ANOVA followed by Tukey's HSD test, $p=3.8\times10^{-9}$; *Figure 7B*). Consistent with our hypothesis, the difference in mean Tolerance values between neurons with and without choice-related activity was greater in the SPC window ($\Delta$Tolerance = 0.12) than in the SD window ($\Delta$Tolerance = 0.04). Second, our hypothesis implies that only neurons with choice-related activity should show a statistically significant increase in Tolerance between the time windows. To test this, we compared the Tolerance values for each neuronal subpopulation across the SD and SPC windows. Indeed, the Tolerance values significantly increased for neurons with choice-related activity ($p=3.8\times10^{-9}$; *Figure 7C*) but not neurons without choice-related activity (p=0.06). These results indicate that the increase in the robustness of 3D selectivity was specific to neurons with choice-related activity. For neurons with choice-related activity, we additionally found that the SPC window Tolerance was significantly correlated with the strength of choice-related activity (log of the choice tuning curve amplitude): r = 0.45, $p=2.8\times10^{-11}$. Thus, more robust 3D selectivity was associated with stronger choice-related activity.

The analysis of Tolerance values indicated that 3D orientation tuning became more similar across distance after the onset of choice-related activity but could not reveal which aspects of the tuning curves (preferences, bandwidths, etc.) specifically changed. We therefore quantified how much the orientation preferences of individual neurons differed across distance in each time window. For each neuron with choice-related activity, we used the orientation preferences from the Bingham function fits in both time windows to calculate a principal orientation preference (see Materials and methods; *Figure 7D*). The deviations in each neuron's orientation preferences from its principal orientation are shown for both time windows in *Figure 7E*. In these plots, the angular variable is the direction that a given preferred orientation deviated from the principal orientation ($\Delta\Psi$) and the radial variable is the angular deviation ($\Delta\zeta$). The origin indicates no difference and the outer ring indicates the maximum difference (90°). Note that the deviations were more tightly clustered around the origin (i.e. smaller deviations) in the SPC window than the SD window. To quantify how much the orientation preferences differed in the two time windows, we calculated cumulative density functions for the angular deviations (*Figure 7F*). The mean deviation was greater in the SD (28.3°) than the SPC (20.4°) window and the cumulative density functions were significantly different between the time windows (Kolmogorov-Smirnov test, $p=1.4\times10^{-10}$). Thus, the orientation preferences of individual neurons were

more similar across distance after the onset of choice-related activity than before. Importantly, we repeated this analysis for neurons without choice-related activity and found that the cumulative density functions were not significantly different across the time windows (p=0.91). These findings indicate that choice-related activity was associated with an increase in the similarity of 3D orientation preferences across distance.

We additionally tested if choice-related activity was associated with changes in the shape of the 3D orientation tuning curves. To do so, we performed a parametric analysis using the Bingham function fits (*Figure 7—figure supplement 1*). First, we found that the tuning bandwidths were not significantly different between neurons with and without choice-related activity in either time window (Kruskal-Wallis test followed by Tukey's HSD test, both p≥0.15). Nor were the bandwidths significantly different across the time windows for either subpopulation (both p≥0.16). Second, we found that the orientation tuning curves of both subpopulations were less elongated (more isotropic) in the SPC window than the SD window (both p≤0.04). For instance, this difference is seen in the orientation tuning curves at 37 cm for the example neuron without choice-related activity (*Figures 3D* and *6D*): SD $\lambda_1 = -1.84$ (elongated) and SPC $\lambda_1 = -0.05$ (almost isotropic). Thus, regardless of choice-related activity, tilt and slant tuning widths were more similar later in the response than earlier. Although the isotropy of the tuning curves was similar for the two subpopulations in the SD window (p=0.89), neurons with choice-related activity had more isotropic tuning than neurons without choice-related activity in the SPC window (p=$1.4 \times 10^{-6}$). Third, we found that the orientation of the major/minor axes of the tuning curves generally aligned with the slant and tilt axes of the coordinate space, and did not differ between the subpopulations in either time window or across time windows (all four p≥0.72). These findings indicate that choice-related activity was associated with a greater change in the aspect ratio of the tuning curves, such that the slant and tilt tuning widths became more similar. One potential consequence of this change is that the Fisher information along the two orientation dimensions would become more similar. This could theoretically make the monkeys' sensitivities to small differences in slant or tilt more equivalent (*Seung and Sompolinsky, 1993*).

## Sensorimotor associations in CIP are strongest for neurons with choice-related activity

We next wanted to test if sensorimotor associations between 3D surface representations and motor-related activity exist at the level of CIP. However, it is currently unknown if CIP carries motor-related activity. Anatomical and effective connectivity studies indicate that CIP receives (direct or indirect) input from V3A and projects to LIP (*Nakamura et al., 2001*; *Premereur et al., 2015*; *Van Dromme et al., 2016*), two areas with saccade-related activity (*Andersen et al., 1992*; *Nakamura and Colby, 2000*). These connections prompted us to test for saccade direction tuning using a center-out, visually guided saccade task (*Munoz and Wurtz, 1995*; *Hanes and Schall, 1996*). In the task, fixation was held on a target at the center of the screen for 1.3 s (to match the total fixation duration in the discrimination task). The fixation target then disappeared and a single saccade target appeared at one of eight locations which coincided with the choice target locations. A saccade was then made to that target (*Figure 8A*).

Importantly, the visually guided saccade task revealed that CIP carries motor-related activity. Following our analysis of choice-related activity, we computed eight population-level time courses of saccade-related activity relative to the saccade direction that elicited the maximum response for each neuron (*Figure 8B*). Saccade direction selectivity started 102 ms before the saccade onset (ANOVA, p<0.05). The responses showed parametric tuning with amplitudes that symmetrically fell off with greater deviation from the preferred direction. They also predicted the saccade timing. For each neuron, we labeled every trial in which a saccade was made in the preferred direction according to the saccade latency. The time course of saccade-related activity was then calculated for each quartile of the saccade latencies (*Figure 8C*; inset shows the latency histogram, mean = 137 ms). The four latency-conditioned time courses approximately intersected 15 ms before saccade onset, suggestive of a saccade initiation threshold (~42 spikes/s, open black circle). The activity also increased more slowly when the saccade latency was longer (colored circles mark when each curve deviated from baseline; ANOVA, p<0.05). For each time course, we computed the growth rate (linear slope) between the start of activity and the putative saccade threshold. Consistent with frontal eye field findings (*Hanes and Schall, 1996*), there was an inverse linear relationship between the

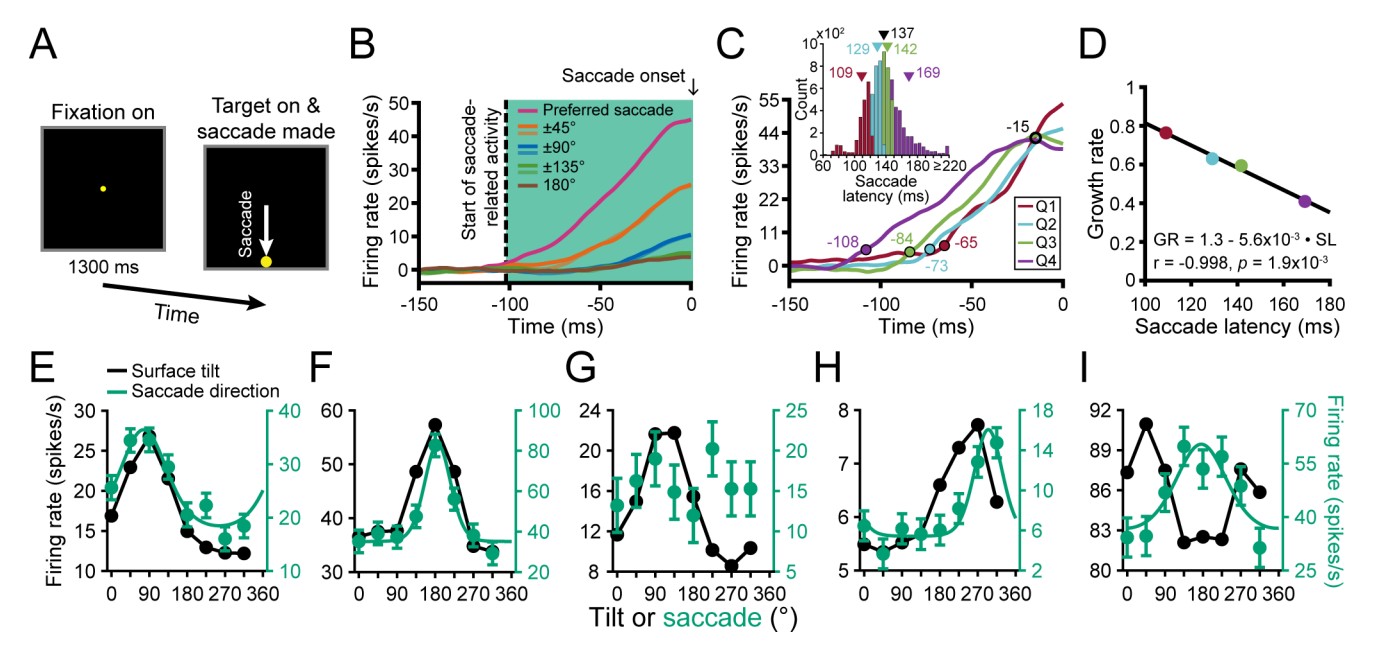

**Figure 8.** Saccade-related activity and sensorimotor associations. (A) Saccade task. A target was fixated for 1.3 s after which it disappeared and a saccade target appeared at one of eight locations. A saccade was then made to that target. (B) Time course of saccade-related activity. Curves show responses averaged over neurons for each of the eight saccade directions relative to each neuron's preferred direction. Saccade onset = 0 ms. Vertical dashed line marks the start of saccade-related activity (−102 ms). Shaded region marks the window used to assess saccade-related activity. (C) Time course of saccade-related activity conditioned on the saccade latency in quartiles (Q). Colored circles mark the start of activity for each quartile. Open black circle marks a putative saccade threshold. Times are indicated next to the circles. Inset shows the histogram of saccade latencies with quartiles colored. Triangles mark mean values (black for the whole distribution). (D) Inverse linear relationship between the growth rate (GR) of saccade-related activity and mean saccade latency (SL) for each quartile. (E-I) Comparison of surface tilt and saccade direction tuning curves for the example neurons (*Figures 3A–E*, *5B–F* and *6A–E*, same order). Black points are surface tilt responses marginalized over slant and distance (SD and SPC windows). Note that the tuning curve in F has a single peak rather than two (c.f., *Figure 5C*), reflecting an increase in the robustness of 3D orientation selectivity across distance. Green points are saccade direction responses and the curves are von Mises fits (tuned cases only).

growth rate and mean saccade latency (*Figure 8D*). Thus, saccade-related activity in CIP functionally correlated with both the saccade direction and timing during the visually guided saccade task.

Across the population, 274 neurons (63%) had significant saccade direction tuning (ANOVA, p<0.05). Saccade direction tuning curves with von Mises fits are shown for the example neurons in *Figure 8E-I* (green curves). The tuning curves were well described by von Mises functions (mean r = 0.91 ± 0.10 Std Dev, N = 274). The mean concentration parameter was $\kappa$ = 4.67 ± 4.84 Std Dev (HWHH: 42° ± 17° Std Dev). For comparison, surface tilt tuning curves marginalized over slant and distance (SD and SPC windows) are also shown (black curves). The first two neurons had orientation tuning (*Figures 3A,B* and *6A,B*), choice tuning (*Figure 5B,C*), and saccade direction tuning (*Figure 8E,F*). The surface tilt and saccade direction preferences of these neurons were well aligned, consistent with a sensorimotor association. The third neuron had orientation tuning (*Figures 3C* and *6C*) and choice tuning (*Figure 5D*) but not saccade direction tuning (*Figure 8G*). The fourth neuron, which had orientation tuning (*Figures 3D* and *6D*) but not choice tuning (*Figure 5E*), had saccade direction tuning (*Figure 8H*). The surface tilt and saccade direction preferences of this neuron were somewhat aligned but not as well as for the neurons with strong orientation and choice tuning. Lastly, the neuron with stronger distance than orientation tuning (*Figures 3E* and *6E*) had choice tuning (*Figure 5F*) and saccade direction tuning (*Figure 8I*). The saccade direction preference of this neuron did not align with its surface tilt or choice preferences (which were aligned).

Having discovered saccade direction selectivity in CIP, we were able to test for a sensorimotor association between the surface tilt and saccade direction preferences. For each neuron with significant saccade direction tuning, we compared the principal surface tilt (SD and SPC windows) and saccade direction preference from the von Mises fit. Across the population, the preferences aligned

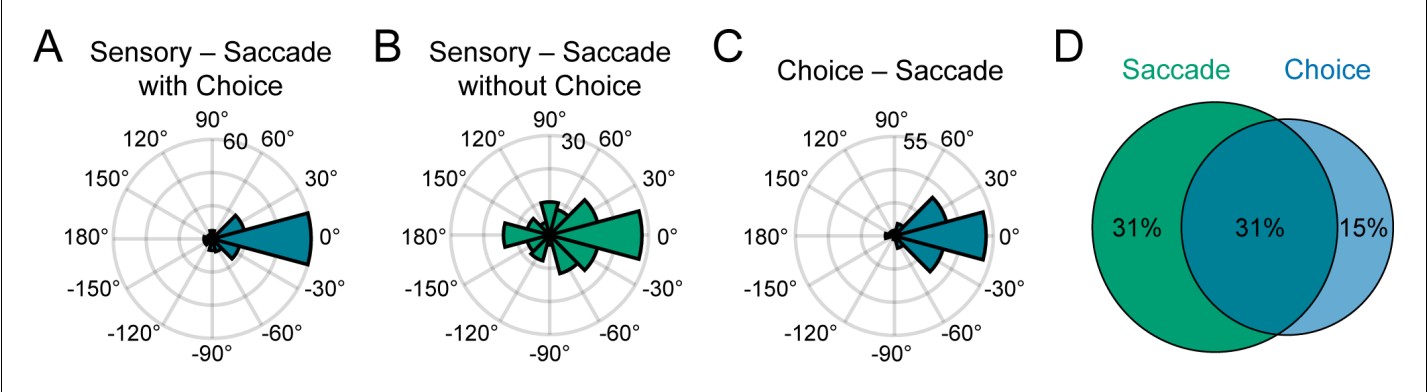

**Figure 9.** Sensorimotor and choice-motor associations. (A) Differences between principal surface tilts and saccade direction preferences for neurons with choice-related activity (N = 135). (B) Same as A, but for neurons without choice-related activity (N = 131). (C) Differences between choice and saccade direction preferences (N = 137). (D) Venn diagram showing the percentages of neurons with saccade-related activity only (green), choice-related activity only (blue), and both (teal).

both for neurons with (*Figure 9A*) and without (*Figure 9B*) choice-related activity, indicating a sensorimotor association. For neurons with (without) choice-related activity, the median circular difference between preferences was −1.6° (−0.9°), and not significantly different from 0° (circular median test, both p≥0.61). Although the tilt and saccade direction preferences generally aligned regardless of choice-related activity, the distribution of preference differences was significantly narrower for neurons with (circular variance = 0.36) than without (circular variance = 0.74) choice-related activity (two-sample concentration difference test, p=3.2×10$^{-5}$; *Fisher, 1995*). Thus, these results indicate that the sensorimotor association (i.e. the alignment of surface tilt and saccade direction preferences) did not require choice-related activity, but was strongest for neurons with choice-related activity, revealing a three-way interaction between sensory representations, choice-related activity, and motor-related activity.

Across the population, 137 neurons (31%) had both choice- and saccade-related activity. For these neurons, the choice and saccade direction preferences generally aligned (*Figure 9C*). The median circular difference between the preferences was 2.9° and not significantly different from 0° (circular median test, p=0.61). Although the preferences tended to align, the differences between the choice and saccade properties of the example neurons suggested that these functional properties were distinct (*Figure 8E-I*). Supporting this distinction, the choice tuning curves were systematically broader than the saccade direction tuning curves (average ΔHWHH = 12°; within-neuron comparison, N = 137), and the difference was statistically significant (sign tests on the von Mises $\kappa$'s and HWHHs, both p≤4.0x10$^{-6}$). We also found that many neurons had saccade-related activity only (137, 31%) or choice-related activity only (64, 15%), indicating that these properties were dissociable across the population (*Figure 9D*). Together, these results suggest that choice- and saccade-related activities were associated but functionally distinct.

### Distinguishing functional links between the robustness of 3D selectivity and choice- versus saccade-related activity

We lastly wanted to assess if the robustness of 3D selectivity was distinctly associated with choice-related activity or if it was also associated with saccade-related activity. To do so, we identified four subpopulations of neurons based on their choice- and saccade-related activities. Specifically, those with: (1) choice- and saccade-related activity (C+S+, N = 137), (2) choice- but not saccade-related activity (C+S-, N = 64), (3) no choice- but saccade-related activity (C-S+, N = 137), and (4) neither choice- nor saccade-related activity (C-S-, N = 99). First, we compared the two subpopulations with choice-related activity (C+S+ and C+S-). The Tolerance values of these subpopulations were not significantly different in either time window (ANOVA followed by Tukey's HSD test, both p≥0.14; *Figure 10A,B*). Likewise, the cumulative density functions for the angular deviations in orientation preferences were not significantly different in either time window (Kolmogorov-Smirnov test, both p≥0.24; *Figure 10C*). Thus, for neurons with choice-related activity, the robustness of 3D selectivity

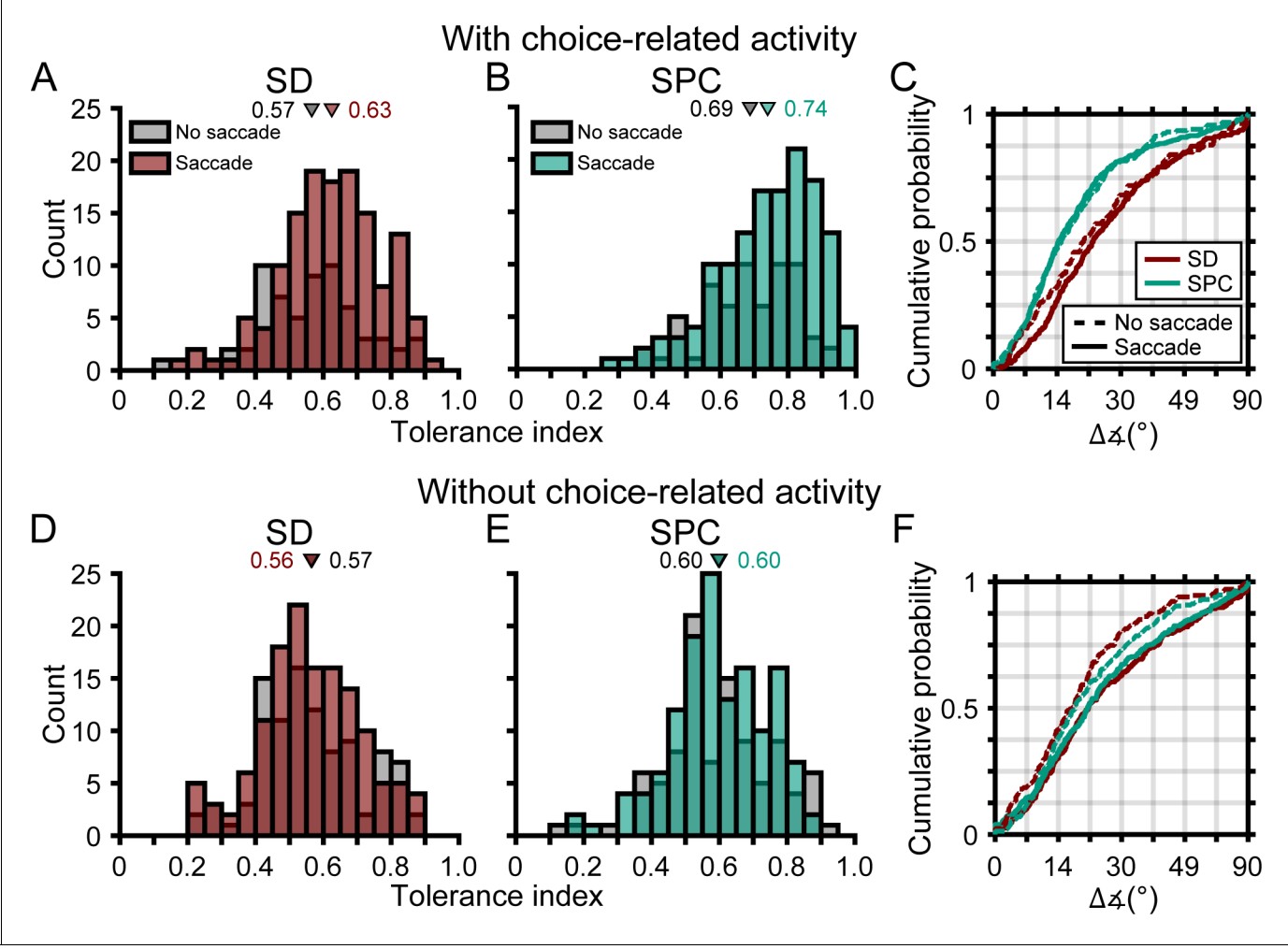

**Figure 10.** The robustness of 3D selectivity was associated with choice-related activity but not saccade-related activity. (**A-C**) Neurons with choice-related activity. (**A**) Tolerance values in the SD window for neurons with (brown) and without (gray) saccade-related activity. (**B**) Tolerance values in the SPC window for neurons with (green) and without saccade-related activity. (**C**) Cumulative density functions for the angular deviations in orientation preferences for neurons with (solid lines) and without (dashed lines) saccade-related activity. (**D-F**) Same as **A-C**, but for neurons without choice-related activity. Triangles in **A,B,D,E** mark mean values.

was not associated with saccade-related activity. Across time windows, both subpopulations showed similar increases in Tolerance (both $p \leq 7.9 \times 10^{-5}$) and similar reductions in the angular deviations in orientation preferences across distance (both $p \leq 0.014$). Thus, choice-related activity was sufficient for the robustness of 3D selectivity to increase across the time windows and saccade-related activity was not necessary. Second, we compared the two subpopulations without choice-related activity (C-S+ and C-S-). The Tolerance values of these subpopulations were not significantly different in either time window (both p=1.0; *Figure 10D,E*). The cumulative density functions for the angular deviations in orientation preferences were significantly different in the SD window ($p=3.8 \times 10^{-3}$; preferences were more similar for neurons without than with saccade-related activity) but not the SPC window (p=0.10; *Figure 10F*). Thus, for neurons without choice-related activity, saccade-related activity was not associated with more robust 3D selectivity. Across time windows, neither subpopulation showed significant changes in Tolerance (both $p \geq 0.56$) or in the angular deviations in orientation preferences across distance (both $p \geq 0.18$). Thus, choice-related activity was necessary for the robustness of 3D selectivity to increase across the time windows and saccade-related activity was not sufficient. These results further support that choice- and saccade-related activities were functionally distinct and suggest that the robustness of 3D visual representations was linked to choice-related activity but not saccade-related activity.

## Discussion

Visually guided interactions with the environment require the creation of robust 3D object representations from ambiguous sensory signals and mapping of those representations to appropriate motor responses. In this study, we investigated the relationships between the robustness of 3D visual representations, choice-related activity, and motor-related activity in area CIP, a relatively understudied region of parietal cortex that was only recently confirmed to be an architectonically distinct brain region (*Katsuyama et al., 2010*; *Niu et al., 2020*). Our findings revealed previously unrecognized links between the fidelity of high-level sensory representations, choice-related activity, and motor-related activity. They further implicate CIP in flexible, experience-dependent sensorimotor transformations that map information about the 3D world into overt behavior.

### Building neuronal selectivity for 3D object pose

Transforming 2D retinal images into 3D visual representations is a nonlinear optimization problem (*Hartley and Zisserman, 2003*). Neurons in CIP achieve 3D orientation selectivity through the integration of stereoscopic and perspective cues (*Tsutsui et al., 2001*; *Tsutsui et al., 2002*; *Rosenberg and Angelaki, 2014b*). A stereoscopic representation of 3D orientation that is tolerant to distance requires an encoding of relative disparity gradients that accounts for the nonlinear relationship between depth and disparity. We found that the orientation selectivity of many CIP neurons was highly tolerant to distance. Importantly, that tolerance did not simply reflect selectivity for perspective cues (which were independent of distance) since the vast majority of neurons also had significant distance tuning. Our findings thus revealed a high-level representation of 3D object pose that was created, at least in part, through relative disparity computations. How might this representation be achieved?

Functional properties and connectivity data suggest that representations of 3D object pose based on relative disparities are created hierarchically, beginning with V1 neurons that represent local absolute binocular disparities (*Cumming and Parker, 1997*; *Cumming and Parker, 1999*; *Cumming and Parker, 2000*). A transformation towards relative disparity tuning proceeds in V2 (*Thomas et al., 2002*) and V3A where absolute disparity representations may still predominate (*Anzai et al., 2011*) but selectivity for disparity gradients may also begin (*Elmore et al., 2019*). The 2D-to-3D transformation likely progresses further in the posterior intraparietal area where tuning for local relative disparity gradients may be solidified (*Alizadeh et al., 2018*), ultimately achieving 3D object pose selectivity in CIP.

In addition to 3D object representations, recent studies have identified brain regions that represent the 3D structure of the environment (*Vaziri et al., 2014*; *Lescroart and Gallant, 2019*). It is currently unclear how 3D object and environment representations interact. To characterize 3D sensory and sensorimotor processing in naturalistic contexts, it will be important for futures studies to investigate how CIP represents objects with which the monkeys directly interact and if that representation is influenced by the surrounding environment.

### Separable tuning for orientation and distance

Distinguishing 3D representations from lower-level visual feature selectivity is a longstanding problem. Demonstrating that 3D orientation selectivity is tolerant to distance when the fixation distance is held constant can be used to verify 3D tuning (*Janssen et al., 2000*; *Nguyenkim and DeAngelis, 2003*; *Alizadeh et al., 2018*; *Elmore et al., 2019*). This fundamental test was not previously performed in CIP. We found that the slant preferences of individual neurons tended to increase with distance behind fixation. Although the increase was relatively small, it indicates that orientation and distance tuning was not strictly separable. Importantly, this finding was consistent with the behavioral data which showed that tilt sensitivity fell off more gradually with distance from fixation at larger compared to smaller slants, which would be predicted if the amplitude of the neuronal responses fell off more slowly with distance at larger compared to smaller slants (*Chang et al., 2020*). These findings suggest that strictly separable 3D pose tuning may not be a reasonable expectation. It would be informative to test if this conclusion rests on the use of stimuli with fixed slants as opposed to fixed disparity gradients. Although the disparity gradient signaling a particular slant is distance dependent (due to the nonlinear relationship between depth and disparity), the 3D shape of a stimulus changes with distance if the disparity gradient is fixed (for the same reason). As

such, neuronal tuning for orientation and distance may not be strictly separable regardless of whether the stimuli have fixed slants or fixed disparity gradients.

## Associations between 3D selectivity and choice-related activity

We found that neurons with more robust 3D representations (i.e. those whose 3D orientation selectivity was more tolerant to distance) preferentially carried choice-related activity. Moreover, greater tolerance was correlated with stronger choice-related activity. Thus, the activity of neurons with more robust 3D selectivity was more tightly coupled to the monkeys' behavior. The extent to which subcortical vestibular neurons resolve the gravito-inertial acceleration ambiguity necessary to represent translation independently of head tilt similarly correlates with the strength of choice-related activity (*Liu et al., 2013*; *Dakin and Rosenberg, 2018*). Across sensory modalities, these findings suggest that neurons which have resolved fundamental ambiguities about the sensory information being discriminated preferentially carry choice-related activity.

Our behavioral task required the monkeys to report the tilt of a plane regardless of its slant or distance. As such, tilt was the directly relevant stimulus feature while slant and distance were nuisance variables that modulated performance (*Chang et al., 2020*). This distinction between the stimulus features allowed us to test for interactions between choice-related activity and the representations of directly relevant and 'nuisance' sensory information. Following the onset of choice-related activity there was no attenuation in selectivity for slant or distance across the population. Thus, the low-dimensional task ('report the tilt') did not reduce the number of features maintained in the neuronal representation of the multi-dimensional (tilt, slant, and distance) stimulus. Instead, for neurons with choice-related activity, the robustness of 3D selectivity increased across the SD and SPC windows. Thus, choice-related activity was associated with an increase in the fidelity of the high-level sensory representations. Because the behavioral task required the monkeys to report the surface tilt only, we were restricted to measuring choice tuning curves over tilt. However, the increased robustness in 3D selectivity cannot be fully explained by the addition of a tilt choice tuning curve at each stimulus distance since choice-related activity was associated with changes in both the tilt and slant preferences (thereby aligning the 3D orientation tuning curves across distance). Thus, we speculate that the changes in 3D selectivity reflected choice-related activity that contained both tilt and slant information. Importantly, the increased fidelity of the 3D representations may enable more robust perception and flexible sensorimotor processing. For example, if choice-related processes attenuated selectivity for features not directly relevant to the task at hand, performance would be impaired if the task and relevant information unexpectedly changed.

Choice-related activity is traditionally associated with feedforward contributions to perception (*Celebrini and Newsome, 1994*; *Britten et al., 1996*; *Dodd et al., 2001*; *Nienborg and Cumming, 2006*; *Gu et al., 2007*). However, this possibility has been questioned on the basis that choice-related activity, attentional effects, and correlated activity can be conflated (*Cumming and Nienborg, 2016*). Indeed, existing measures of choice-related activity likely reflect combinations of these factors (*Haefner et al., 2013*; *Gu et al., 2014*). We found that surface tilt and choice preferences typically aligned, consistent with a feedforward origin of choice-related activity and contrasting with the potentially non-specific effects of a feedback origin, as thought to occur in the ventral intraparietal area (*Zaidel et al., 2017*). Thus, a potential implication of the current findings is that neurons with more robust tuning for task-relevant information have greater weight in the neural readout. Indeed, decoding from neurons that are more tolerant to nuisance variables may simplify readout since marginalizing out non-relevant features would be more straightforward.

For the current study, we trained monkeys to report the near side of a viewed plane, whereas a previous CIP study had monkeys report the far side (*Elmore et al., 2019*). Thus, the monkeys in these two studies were trained to make opposite behavioral reports for the same surface tilt. This difference allowed us to test if the alignment of surface tilt and choice report directions was experience dependent. In both data sets we found that the alignment of preferred surface tilt and choice report directions reflected the task training. Importantly, this collective finding indicates that the association between 3D visual orientation and choice report directions did not simply reflect a natural correspondence. Instead, it was flexible, reflecting an experience-dependent association between perception and action.

## Sensorimotor and choice-motor associations

We discovered that the functional properties of CIP neurons extend beyond sensory and perceptual processing. Specifically, CIP activity correlated with the direction and timing of visually guided saccades, implicating the area in visuomotor control for the first time. Moreover, there was a sensorimotor association between the surface tilt and saccade direction preferences, indicating that visual and oculomotor activities were not simply multiplexed. Correspondingly, for neurons with both choice- and saccade-related activity, the choice report and saccade direction preferences also generally aligned. However, it is important to emphasize that the choice- and saccade-related activities were distinct from each other. First, the choice and saccade tuning curves of individual neurons systematically differed in bandwidth. Second, these functional properties were dissociable since many neurons were only tuned for choice report or saccade direction, not both. These findings suggest that the choice-related activity did not simply reflect saccade preparation signals. Moreover, the dissociation of these properties across the population allowed us to test if the robustness of 3D selectivity was distinctly associated with choice-related activity or if it was also linked to saccade-related activity. For example, it is conceivable that the addition of saccade preparation signals to the sensory responses might appear like an increase in the robustness of 3D selectivity since the saccade signals in our task were likely independent of the surface's distance. However, that was not the case. Across the population, we found that choice-related activity was both necessary and sufficient for the robustness of 3D selectivity to increase across the time windows. In contrast, saccade-related activity was neither necessary nor sufficient. If choice- and saccade-related activities simply reflected a common signal, then saccade-related activity would have also likely been associated with an increase in the fidelity of the 3D visual representations, which was not the case. Thus, these findings further support that choice- and saccade-related activities were functionally distinct, and suggest that only choice-related activity was linked to the robustness of the 3D visual representations.

We further found that sensorimotor associations were strongest for neurons with choice-related activity (and therefore the most robust 3D selectivity). Thus, a potential implication of the current findings is that neurons with robust 3D selectivity and strong sensorimotor associations have the greatest weight in determining motor responses. Indeed, associating robust sensory and motor-related activity at the single neuron level may help ensure successful and timely interactions with the environment since the same neurons represent the relevant sensory information and signal the appropriate motor response. Building on these findings, future studies can test this possibility using reaction time tasks to evaluate the time course of 3D sensory and sensorimotor processing. To date, relatively few studies have used reaction time tasks to assay 3D visuomotor processing (*Verhoef et al., 2012*; *Verhoef et al., 2015*). Such tasks could elucidate potential differences in the timing of choice-related activity across neurons, sessions, and/or conditions, which this study was not designed to examine, and the implications of such differences for 3D perception and action.

In addition to oculomotor processing, connections between CIP and prehensile areas suggest that the area's sensorimotor functions may be broader. This possibility is consistent with recent work implicating human parietal regions that integrate visual orientation and saccade signals in the updating of grasp plans (*Baltaretu et al., 2020*), and will be important to explore through neurophysiological studies with monkeys. Lastly, our findings reveal that experience has ongoing effects on the functional properties of CIP neurons, as occurs in downstream area LIP (*Freedman and Assad, 2006*; *Law and Gold, 2008*; *Bennur and Gold, 2011*), producing flexible mappings between sensory, choice-, and motor-related activity that support goal-directed behaviors. Determining how such effects influence downstream motor processing will be essential to further elucidate the sensorimotor cascade underlying ecologically relevant interactions with the 3D environment.

## Materials and methods

### Animal preparation

This study was performed in strict accordance with the recommendations of the National Institutes of Health's Guide for the Care and Use of Laboratory Animals. All experimental procedures and surgeries were approved by the Institutional Animal Care and Use Committee (IACUC) at the University of Wisconsin–Madison (Protocol G005229). Two male rhesus monkeys (*Macaca mulatta*; Monkey L: 6 years of age, ~9.5 kg; Monkey F: 5 years, ~6.4 kg) were surgically implanted with a Delrin ring for

head restraint and a removable recording grid for guiding electrodes (*Rosenberg et al., 2013*). After recovery, the monkeys were trained to sit in a primate chair with head restraint and to fixate visual targets within 2° version and 1° vergence windows.

## Experimental control and stimulus presentation

Experimental control was performed using the open-source REC-GUI software (RRID:SCR_019008; *Kim et al., 2019*). Stimuli were rendered using Psychtoolbox 3 (MATLAB R2016b; NVIDIA GeForce GTX 970). They were rear-projected onto a polarization preserving screen (Stewart Film Screen, Inc) using a DLP LED projector (PROPixx; VPixx Technologies, Inc) with 1280 × 720 pixel resolution (70° × 43°) at 240 Hz (120 Hz/eye). The screen was positioned 57 cm from the monkey. Polarized glasses were worn. A phototransistor circuit was used to confirm the synchronization of left and right eye images and to align neuronal responses to the stimulus onset. Eye positions were monitored optically at 1 kHz (EyeLink 1000 plus, SR Research).

## Visual stimuli

The stimuli were the same as the combined-cue stimuli in our previous work (*Chang et al., 2020*). They consisted of 250 nonoverlapping dots uniformly distributed across a plane. The stimulus envelope was a 20° diameter circle on the screen. The background was gray (11.06 cd/m$^2$) and the dots were bright (35.1 cd/m$^2$), measured through the glasses (PR-524 LiteMate, Photo Research). Planes were presented at all combinations of eight tilts (0° to 315°, 45° steps), five slants (0° to 60°, 15° steps), and four distances (37, 57, 97, and 137 cm). Fixation was always at screen distance (57 cm). At 37 cm, all dots were in front of the fixation target. At 57 cm, the dots were distributed about the fixation target. At 97 and 137 cm, all dots were behind fixation. Presenting the stimuli in front of, distributed about, and behind fixation prevented the monkeys from relying on local absolute disparity cues to perform the task. The dots were rendered with stereoscopic and perspective cues and scaled according to the distance so that their screen size only depended on slant. At S = 0°, the dot size was 0.35° isotropic.

## Tilt discrimination task

The monkeys were trained to perform an 8AFC tilt discrimination task (*Chang et al., 2020*). Each trial began with fixation of a circular target (0.3°) at the center of the screen for 300 ms. A plane then appeared at the center of the screen for 1 s. The target and plane then disappeared and eight choice targets appeared at polar angles of 0° to 315° in 45° steps (11° eccentricity). The nearest side of the plane was reported by making a saccade to the corresponding target for a liquid reward. Responses to frontoparallel planes (slant = 0°, tilt undefined) were rewarded with equal probability (12.5%).

## Visually guided saccade task

Each trial began with fixation of a target at the center of the screen for 1.3 s to match the total fixation duration in the 8AFC tilt discrimination task. The fixation target then disappeared and a single saccade target appeared at one of the eight choice target locations. A saccade to the target was made for a liquid reward.

## Experimental protocol

Stimuli were presented in a pseudorandom order within a block design. A block included one completed trial for each of the following: (*i*) planes: (8 tilts × four non-zero slants + eight frontoparallel trials) × four distances (N = 160), and (*ii*) saccades: 8 directions × four repeats (N = 32). A trial was aborted and data discarded if fixation was broken before the choice or saccade targets appeared, or if the trial was not completed within 500 ms of their appearance. At least five completed blocks were required to include a neuron for analysis.

## Neuronal recordings

Area CIP was identified based on anatomical and functional properties (*Rosenberg et al., 2013*; *Rosenberg and Angelaki, 2014a*; *Rosenberg and Angelaki, 2014b*; *Elmore et al., 2019*). Briefly, the CARET software was used to register the structural MRIs to the F99 atlas (*Van Essen et al.,*

*2001*) and align the recording grids to estimate electrode trajectories. The area was functionally distinguished from neighboring regions based on 3D orientation selectivity and large receptive fields. Recordings were performed using silicone linear array probes with either four or eight tetrodes (NeuroNexus, Inc). Tetrodes were separated by 300 µm. Electrodes within a tetrode were arranged in a diamond pattern and separated by 25 µm. Neuronal signals were sampled at 30 kHz and stored with eye movement and phototransistor traces sampled at 1 kHz (Scout Processor; Ripple, Inc). Tetrode-based spike sorting was performed offline using the KlustaKwik (K. Harris) semi-automatic clustering algorithm in *MClust* (MClust-4.0, A.D. Redish et al.) followed by manual refinement using Offline Sorter (Plexon, Inc). Only well-isolated single neurons verified by at least two authors were included for analysis: 218 neurons from the left hemisphere of Monkey L (26 sessions) and 219 neurons from the right hemisphere of Monkey F (27 sessions).

## Analysis of the behavioral data

Tilt discrimination performance was quantified by fitting a von Mises probability density function to the distribution of reported tilt errors (*Chang et al., 2020*), ΔTilt = reported tilt – presented tilt:

$$VM(\Delta\mathrm{Tilt}) = e^{\kappa \cdot \cos(\Delta\mathrm{Tilt}-\mu)}/(2\pi \cdot I_0(\kappa)). \tag{1}$$

The mean ($\mu$) and concentration ($\kappa$) describe accuracy and sensitivity, respectively (*Dakin and Rosenberg, 2018*; *Seilheimer et al., 2014*). Values of $\mu$ closer to 0 indicate greater accuracy. Larger $\kappa$ indicate greater sensitivity. Given the 45° tilt sampling interval, we set $\kappa = 18$ as the upper bound in the estimation routine (*Chang et al., 2020*). A modified Bessel function of order 0, $I_0(\kappa)$, normalizes the function to have unit area.

## Analyses of the neuronal data

### Visual response latency

To estimate a neuron's visual response latency, a spike density function (SDF) was created for each trial that a plane was presented by convolving the spike train (1 ms bins, aligned to the stimulus onset) with the function (*Schwemmer et al., 2015*):

$$\alpha(t) = \alpha^* \cdot H(t) \cdot [e^{-\tau_d \cdot t} - e^{-\tau_r \cdot t}]. \tag{2}$$

Here, $\alpha^*$ normalizes $\alpha(t)$ to have unit area, $H(t)$ is the Heaviside function, and $\tau_d = 0.05$ and $\tau_r = 1.05$ are time constants. The response latency was defined as the first time point after stimulus onset that the firing rate was significantly different from baseline for at least 30 ms (ANOVA, $p<0.05$). The baseline firing rate was calculated using the last 150 ms of the fixation periods preceding the stimulus onsets.

### Tilt discrimination index (TDI)

We computed a TDI to quantify how well preferred and non-preferred tilts could be discriminated using individual neuron responses (*Prince et al., 2002*; *Elmore et al., 2019*):

$$\mathrm{TDI} = \frac{R_{max} - R_{min}}{R_{max} - R_{min} + 2\sqrt{SSE/(N-M)}}. \tag{3}$$

Here, $R_{max}$ and $R_{min}$ are the maximum and minimum mean responses across a tilt tuning curve, $SSE$ is the sum squared error around the mean responses, $N$ is the total number of trials, and $M$ is the number of tilts ($M = 8$). A TDI was calculated for each combination of slant and distance. Neurons with strong (weak) response modulation relative to their variability have TDIs closer to 1 (0).

### Choice-related activity

We tested for choice-related activity using the frontoparallel plane data. To combine data across distances, we z-scored all baseline subtracted frontoparallel plane responses at each distance. We then grouped the z-scored firing rates according to the choice. A neuron was classified as having choice-related activity if the z-scored firing rates significantly depended on the choice (ANOVA, $p<0.05$). For each of these neurons, an average SDF was calculated for each choice (aligned to the stimulus onset), and labeled relative to the choice that elicited the maximum z-scored firing rate for the

neuron: preferred choice, ±45°, ±90°, ±135°, and 180°. The SDFs were then averaged across neurons to create eight population-level time courses. The onset of choice-related activity was defined as the first time point that the time courses significantly differed (ANOVA, p<0.05). To refine the estimate, we iteratively repeated this process with all neurons, each time calculating the firing rates starting from the previous estimate of the onset of choice-related activity to the end of the 1 s stimulus presentation (firing rates were first calculated using the full 1 s). This process was repeated until the onset no longer changed.

## Quantifying orientation selectivity

For each distance and time window, we tested for orientation tuning (ANOVA, p<0.05 and Bonferroni-Holm corrected). The preferred tilt and slant was estimated for each significant case by fitting a Bingham function (*Rosenberg et al., 2013*). Differences in orientation preference were assessed as follows. First, we calculated the principal orientation about which the measured orientation preferences clustered (*Fisher et al., 1993*). For each distance and time window with significant tuning (N $\leq$ 8), the surface normal vector $n_i = [x_i \ y_i \ z_i]^T$ of the plane with the preferred tilt ($T_i$) and slant ($S_i$) was calculated:

$$
\begin{aligned}
x_i &= \cos(T_i) \cdot \sin(S_i), \\
y_i &= \sin(T_i) \cdot \sin(S_i), \\
z_i &= \cos(S_i).
\end{aligned}
\tag{4}
$$

The normal vectors were then arranged in a matrix and the eigenvectors were calculated. The principal orientation was defined as the eigenvector with the greatest eigenvalue. Principal orientations were calculated using the SD window only and both (SD and SPC) windows. Second, we rotated the normal vectors such that the principal orientation aligned with the north pole ($n = [0 \ 0 \ 1]^T$) and their relative orientations were preserved:

$$
n_i' = rT(T_p) \cdot rS(-S_p) \cdot rT(-T_p) \cdot n_i.
\tag{5}
$$

Here, $n_i'$ is a rotated normal vector, $T_p$ and $S_p$ are the tilt and slant of the principal orientation, respectively, and $rT$ and $rS$ are rotation matrices:

$$
rT(T_p) = \begin{bmatrix} \cos(T_p) & -\sin(T_p) & 0 \\ \sin(T_p) & \cos(T_p) & 0 \\ 0 & 0 & 1 \end{bmatrix}
\tag{6}
$$

and

$$
rS(S_p) = \begin{bmatrix} \cos(S_p) & 0 & \sin(S_p) \\ 0 & 1 & 0 \\ -\sin(S_p) & 0 & \cos(S_p) \end{bmatrix}.
\tag{7}
$$

The equatorial projection of the rotated normal vectors is a standardized polar coordinate space that describes deviations in orientation preference.

## Quantifying tolerance

To quantify how tolerant each neuron's orientation selectivity was to distance, we fit the 33 × 4 (orientations × distances) matrix of responses with a multiplicatively separable model:

$$
R(\theta, D) = DC + g \cdot H(\theta) \cdot F(D).
\tag{8}
$$

Here, $R(\theta, D)$ is the response to orientation $\theta$ (tilt and slant) and distance $D$, $DC$ is an offset, $g$ is the response amplitude, $H(\theta)$ is the orientation tuning, and $F(D)$ is the distance tuning. Fitting was performed using singular value decomposition within a minimization routine to find the $DC$, $g$ (first singular value), and $H$ and $F$ (first pair of singular vectors) that minimized the Euclidean norm of the error. A Tolerance index was defined as the average Pearson correlation between the observed and

model orientation tuning curves at each distance. Tolerance values closer to 1 (0) indicate that the orientation selectivity was more (less) tolerant to distance.

We also tested an additively separable model, $R(\theta, D) = DC + H(\theta) + F(D)$. To fit this model, we constructed a system of equations for the 132 orientation and distance combinations. The solution to this linear regression problem was found by including a regularization parameter that minimized the Euclidian norm of the error (*Hastie et al., 2009*), giving the additive model one more free parameter than the multiplicatively separable model. Results with the additive model were not presented because the responses of every neuron in both time windows (874 comparisons) were better described by the multiplicatively separable model.

### Saccade analysis

A saccade onset was defined as the first time point that the velocity of either eye was $\geq 150°/s$. A neuron was classified as having saccade direction selectivity if the baseline subtracted firing rates significantly depended on the saccade direction (ANOVA, p<0.05). For each of these neurons, an average SDF was calculated for each saccade direction (aligned to the saccade onset) and labeled relative to the saccade direction that elicited the maximum firing rate for the neuron. The SDFs were then averaged across neurons to create eight time courses. The start of saccade-related activity was defined as the earliest time point before saccade onset that the time courses significantly differed (ANOVA, p<0.05). To refine the estimate, we iteratively repeated this process with all neurons, each time calculating the firing rates from the previous estimate of the start of saccade-related activity to the saccade onset. This process was repeated until the onset no longer changed.

### Vergence

Vergence eye movements can potentially contribute to neuronal responses. We therefore tested if our 3D pose tuning measurements were significantly affected by small vergence eye movements that did not violate the vergence window. For each neuron, we performed an ANCOVA to test for main effects of stimulus tuning with vergence included as a covariate (*DeAngelis and Uka, 2003*; *Elmore et al., 2019*). Tilt (linearized into cosine and sine components), slant, and distance were independent factors and the mean vergence over the analyzed time window was a covariate. Both the SD and SPC windows were included because the results were similar for the individual time windows. Only 40 neurons (9%) showed a statistically significant effect of vergence (p<0.05). Moreover, the significance of the stimulus main effects was unchanged for all but 11 neurons (3%) when vergence was included as a covariate. These effects are smaller than those previously reported for disparity tuning in the middle temporal area (*DeAngelis and Uka, 2003*), and indicate that vergence had a minimal impact on the measurements.

## Acknowledgements

We thank L Caitlin Elmore for recording the 'report far' neurons and help with *Figure 5—figure supplement 2* and Satchal Postlewaite for help with spike sorting. This work was supported by the Alfred P Sloan Foundation (FG-2016–6468), Whitehall Foundation (2016-08-18), Greater Milwaukee Foundation (Shaw Scientist Award), and the National Institutes of Health (EY029438). Further support was provided by National Institutes of Health Grant P51OD011106 to the Wisconsin National Primate Research Center and the Eunice Kennedy Shriver National Institute of Child Health and Human Development core grant U54 HD090256 to the Waisman Center.

## Additional information

### Funding

| Funder | Grant reference number | Author |
| --- | --- | --- |
| Alfred P. Sloan Foundation | FG-2016-6468 | Ari Rosenberg |
| Whitehall Foundation | 2016-08-18 | Ari Rosenberg |
| Greater Milwaukee Foundation | Shaw Scientist Award | Ari Rosenberg |

| National Institutes of Health | EY029438 | Ari Rosenberg |

The funders had no role in study design, data collection and interpretation, or the decision to submit the work for publication.

## Author contributions

Ting-Yu Chang, Raymond Doudlah, Conceptualization, Data curation, Software, Formal analysis, Validation, Investigation, Visualization, Methodology, Writing - original draft, Writing - review and editing; Byounghoon Kim, Conceptualization, Resources, Data curation, Software, Formal analysis, Supervision, Validation, Investigation, Methodology, Writing - review and editing; Adhira Sunkara, Conceptualization, Software, Formal analysis, Validation, Investigation, Visualization, Methodology, Writing - original draft, Writing - review and editing; Lowell W Thompson, Formal analysis, Writing - review and editing; Meghan E Lowe, Data curation, Formal analysis, Validation, Writing - review and editing; Ari Rosenberg, Conceptualization, Resources, Data curation, Software, Formal analysis, Supervision, Funding acquisition, Validation, Investigation, Visualization, Methodology, Writing - original draft, Project administration, Writing - review and editing

## Author ORCIDs

Ting-Yu Chang ⓘ https://orcid.org/0000-0003-3964-0905
Raymond Doudlah ⓘ https://orcid.org/0000-0003-3631-5947
Byounghoon Kim ⓘ http://orcid.org/0000-0001-7159-5134
Ari Rosenberg ⓘ https://orcid.org/0000-0002-8606-2987

## Ethics

Animal experimentation: This study was performed in strict accordance with the recommendations of the National Institutes of Health's Guide for the Care and Use of Laboratory Animals. All experimental procedures and surgeries were approved by the Institutional Animal Care and Use Committee (IACUC) at the University of Wisconsin-Madison (Protocol #: G005229).

## Decision letter and Author response

Decision letter https://doi.org/10.7554/eLife.57968.sa1
Author response https://doi.org/10.7554/eLife.57968.sa2

# Additional files

## Supplementary files

• Transparent reporting form

## Data availability

All data generated or analyzed during this study are included in the manuscript and supporting files.

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
