## [Decision Letter]

**Acceptance summary:**

This study advances our understanding of 3D visual representations in the primate posterior parietal cortex, and how encoding of sensory, choice, and motor factors interact in the caudal intraparietal area during perceptual decisions based on visual depth cues.

**Decision letter after peer review:**

Thank you for submitting your article "Choice activity stabilizes sensory representations and mediates sensorimotor associations in parietal cortex" for consideration by *eLife*. Your article has been reviewed by three peer reviewers, one of whom is a member of our Board of Reviewing Editors, and the evaluation has been overseen by Joshua Gold as the Senior Editor. The reviewers have opted to remain anonymous.

The reviewers have discussed the reviews with one another and the Reviewing Editor has drafted this decision to help you prepare a revised submission.

As the editors have judged that your manuscript is of interest, but as described below that additional analysis, substantial revisions to the manuscript, and possibly new data are needed to address a number of major concerns and points of confusion. The reviewers were unanimous in wanting to give you the opportunity to respond to their concerns but also emphasized that substantial improvements are needed to merit acceptance in our journal. We would like to draw your attention to changes in our revision policy that we have made in response to COVID-19 (https://elifesciences.org/articles/57162). First, because many researchers have temporarily lost access to the labs, we will give authors as much time as they need to submit revised manuscripts. We are also offering, if you choose, to post the manuscript to bioRxiv (if it is not already there) along with this decision letter and a formal designation that the manuscript is "in revision at *eLife*". Please let us know if you would like to pursue this option. (If your work is more suitable for medRxiv, you will need to post the preprint yourself, as the mechanisms for us to do so are still in development.)

Summary:

This is an interesting study which examines the role of parietal cortex area CIP in judgments and choices with 3D stimuli. The stimuli consist of a parametric set of patterns at a range of tilts, slants and distances in depth. The monkeys' task is to report the perceived tilt by making a saccade to a target that is closest to the viewer in depth. Neuronal mechanisms underlying 3D perception and particularly choices/decision are relatively understudied, so this study is novel in that regard. The mapping of the tilt to saccade is fixed so that the choice (saccade direction) is known as soon as the stimulus is viewed. The main result of the study is that there is substantial encoding correlated with 3D tilt as well as with choice (saccade direction). Interestingly there is a strong correspondence observed between tilt and saccade direction, which the authors interpret as associative encoding which combined stimulus tilt encoding with encoding of the appropriate saccade direction, suggestive of a role for CIP in mediating decisions during this task.

Essential revisions:

1) All reviewers raised serious concerns about the ability to dissociate stimulus from choice/saccade encoding. This problem is in part due to the experimental design, which uses a rigid link between the monkeys' decisions and the direction of the saccades. Further, there is concern about the attempt to dissect sensory and choice information using different time windows. Sensory only (SO) window and sensory plus choice (SPC) windows are used for analysis. Since the animal can plan his response as soon as he determines the tilt, how can you be sure that the SO window doesn't contain any choice information? More description and rationale for this is needed, and likely a revised analysis approach is needed. How confident are the authors in the determination of the onset of choice information? This could vary from session to session or from neuron to neuron in principle, yet the timings were determined from an across session average. The SO vs SPC time windows are described in a confusing way-as if they truly dissociate sensory and choice information in a clean way. Instead it is an arbitrary temporal cut-off and the windows each likely include a mixture of sensory and choice information.

2) It is potentially of strong interest that there is a correspondence between tilt and choice selectivity as shown in 5G. A key question is whether this correspondence arose because of training, or because this could be a naturally occurring correspondence. It is possible that eye movements are naturally drawn to the closest part of a tilted 3D object. It would be very interesting to test in naïve animals whether this is indeed the case (perhaps during passive presentation of tilted objects and free viewing). If there is no such effect in naïve animals, but it is observed in the trained animals, this would support the idea that the correspondence arose because of training. Was the tuning assessed for each neuron during passive viewing outside the context of the task? If so, this might be a way to get at the question by comparing task vs passive viewing.

3) The reviewers agreed with one another that the results are described throughout the manuscript in a way which implies a causal relationship between CIP activity/encoding and choice. But given the correlational approaches used here, this message should be revised and clarified in order to avoid confusion and to better bring the discussion of the paper in line with the observed results and experimental design.

4) All reviewers were in agreement that the writing of the manuscript needs to be improved substantially in order to help readers follow the logic of the analyses, experimental design, and results.

5) Analysis related to Figure 4: authors should clarify which period of the task they are describing here, is it SO or SPC? They separate sensory (SO) and choice response (SPC) periods. The distinction between SO and SPC seems pretty arbitrary. The only neuronal activity they show up to Figure 5 are heat map. It would be helpful to see the actual time course of the neuronal response for example neurons.

6) Although they show that 5 example neurons, they do not provide clear population analyzes of 3D orientation selectivity. At least, they should show the distribution of tolerance index for all the neurons. Given the response shown in Figure 5A, I wonder how neurons could be tilt selective during the SO period. I'd like to see the actual response of these neurons.

7) There is confusion regarding the discussion of vergence, raised by reviewer 3, which needs to be addressed. Also, the correlation of tilt selectivity between SO and SPC shown in Figure 7 needs to be clarified and explained in more detail.

[Editors' note: further revisions were suggested prior to acceptance, as described below.]

Thank you for resubmitting your work entitled "Functional links between sensory representations, choice activity, and sensorimotor associations in parietal cortex" for further consideration by *eLife*. Your revised article has been evaluated by Joshua Gold (Senior Editor) and a Reviewing Editor.

The manuscript has been improved but there are some remaining issues that need to be addressed before acceptance, as outlined below:

1) The new analysis with previous data required additional clarification and explanation, and perhaps additional analysis. The reviewers agreed that there is concern about whether the observed results can be explained by saccade preparation. Specifically one reviewer states that "I don't understand how 3D selectivity and choice can be aligned as shown in Figure 5 considering that choice changes depending on the training (Figure 5—figure supplement 1). If a neuron is tuned to right forward, and monkey is trained to choose forward, the neuron will fire strongly during SPC when stim is right forward and therefore when monkey is planning a saccade to the right. When trained to report away, the same neuron should, according to Figure 5—figure supplement 1, responds strongly during SPC when planning a saccade to the left. How is that possible given that 3D selectivity is getting stronger during SPC and that 3D selectivity and saccade spatial selective strongly overlap (Figure 8)? I understand that they are not analyzing the same neurons, but then, does it only mean that choice selectivity in CIP reflects saccade preparation? I'm genuinely afraid I am missing something important here."

Reviewer #1:

In reviewing this revised manuscript, the authors have done a thorough job in responding to the previous round of review. They have clarified key points, and in particular largely resolved major concerns about whether the observed association like effects were indeed due to training by including an additional supplementary figure, and by more thoroughly referring to previous results from their group which help address this concern.

Reviewer #2:

The authors did an excellent job in revising the manuscript. Specifically, the comparison with the task where the monkeys had to indicate the opposite features of the stimulus is crucial to demonstrate that the association between stimulus orientation and choice direction was indeed learned. I have no further major comments.

Reviewer #3:

I recognize authors spend a lot of energy trying to address our comments. I am not sure they fully manage to address them though.

Here are the main claims (Abstract):

Choice related activity (saccade) is carried by neurons with robust 3D selectivity.

Following onset of choice activity 3D robustness increased

3D selectivity and saccade selectivity aligned

Here are my main concerns:

1) Specifically, their analysis consisting of testing for choice related activity in the SD window is not convincing. Our point was that some choice related activity could start before the end of SD window and before the start of the SPC window, not that the activity during the entire SD window was related to choice. In this regard, they failed to address our concerns.

2) I am still very confused by Figure 5—figure supplement 2 and the claim that 3D selectivity and saccade/choice selectivity are aligned:

Let's consider a neuron tuned to top near. We have 4 conditions based on task design (near-CN-or far-CF) and stimulus slant (Top Near-TN-or Bottom near-BN): a) CN/TN, b) CN/BN, c) CF/TN, d) CF/BN.

Condition a: choose near with stimulus top near: this neuron will have a strong visual response which will stay sustained since it is aligned to motor plan.

Condition b: Choose near with stimulus bottom near: this neuron will have a weak visual response, which will stay weak during motor planning.

Condition c: choose far with stimulus top near: there will be a strong visual response (similar to a, neuron's favorite stim) which should decrease during motor planning.

Condition d: choose far with stimulus bottom near (saccade to the top): there will be a weak visual response (similar to b, neuron's less favorite stim) which should increase during motor planning.

I am confused how this is translated to Figure 5 sp2 and how does that match authors' claim that "Following onset of choice activity 3D robustness increased" and that choice and "3D selectivity and saccade selectivity are aligned". Separating between SC and SPC window does not help at all to understand the neuronal response properties.

3) Their attempt to dissociate choice and motor planning is not successful. Stimuli and motor action are still strongly linked whether monkeys have to report near or far side. I don't see how it breaks the link.

4) Despite their effort to improve the flow of argument in the article, I still find it very difficult to follow.

---

## [Author Response]

Essential revisions:1) All reviewers raised serious concerns about the ability to dissociate stimulus from choice/saccade encoding. This problem is in part due to the experimental design, which uses a rigid link between the monkeys' decisions and the direction of the saccades.

To dissociate stimulus from choice/saccade encoding, we now compare results when the monkeys were trained to report opposite features of the same stimuli (i.e., the “near” vs. “far” side of the plane). This task difference allows us to dissociate stimulus orientation from choice/saccade directions since opposite choice report directions were made for the same surface orientation. Importantly, the association between stimulus orientation and choice report direction was reversed across the two data sets, indicating that the association reflected task training. For instance, consider a typical neuron preferring bottom-near planes. If the monkey were trained to report the near side, the neuronal responses were larger if the monkey selected the lower choice target. However, if the monkey were trained to report the far side, the responses were larger if the monkey selected the upper choice target. This indicates that the stimulus and choice/saccade encodings were dissociable. These findings are presented in subsection “Choice-related activity was parametrically tuned and aligned with the tilt preferences”, in new Figure 5—figure supplement 2, and further discussed in subsection “Associations between 3D selectivity and choice-related activity”. We provide greater detail about these additions in Essential Revision 2.

Further, there is concern about the attempt to dissect sensory and choice information using different time windows. Sensory only (SO) window and sensory plus choice (SPC) windows are used for analysis. Since the animal can plan his response as soon as he determines the tilt, how can you be sure that the SO window doesn't contain any choice information? More description and rationale for this is needed, and likely a revised analysis approach is needed. How confident are the authors in the determination of the onset of choice information? This could vary from session to session or from neuron to neuron in principle, yet the timings were determined from an across session average. The SO vs SPC time windows are described in a confusing way-as if they truly dissociate sensory and choice information in a clean way. Instead it is an arbitrary temporal cut-off and the windows each likely include a mixture of sensory and choice information.

One of our primary goals was to assess if the robustness of 3D sensory representations was associated with choice-related activity. To test this, we identified two subpopulations of neurons based on whether or not they carried choice-related activity and two time windows based on the onset of choice-related activity. By comparing the 3D tuning properties of neurons between the subpopulations as well as within the subpopulations across the time windows, we were able to identify relationships between the robustness of 3D selectivity and choice-related activity. To help clarify this, we expanded the description and rationale behind the analyses and made it more explicit that the windows were operationally defined based on the population activity (subsection “The representation of 3D surface pose by CIP neurons”, subsection “Choice-related activity was parametrically tuned and aligned with the tilt preferences”).

The reviewers raise a great point about confidence in the onset of choice-related activity. To address this question, we incorporated several new analyses which verified that the two windows defined functionally distinct response epochs at the individual neuron level. We provide greater details in Essential Revision 5, but briefly we: (1) verified the absence of significant choice-related activity in the SO window for individual neurons, (2) showed that the results were highly similar if choice tuning was assessed using full and short SPC windows, thus indicating that the two-window analysis was sufficient to characterize the findings, and (3) verified the time course of choice-related activity using a state-space analysis that provided a second estimate of the onset of choice-related activity. The second estimate of choice onset was similar to the first and corroborated the absence of significant choice-related activity in the SO window. These results are presented in subsection “Choice-related activity was parametrically tuned and aligned with the tilt preferences” and in new Figure 5—figure supplement 1.

While the new analyses confirmed that choice-related activity was not detectable before ~202– 232 ms, we changed the name of the “sensory only (SO)” window to the “sensory dominant (SD)” window (52 to 202 ms) to leave room for the possibility that some choice-related activity may exist in this time window (though it was not detectable). From this point forward in the responses, we refer to the old SO window by its new name (SD window).

We found that the two-window analysis facilitated our goal of testing whether high-level sensory representations were associated with choice-related activity. These findings motivate and will be essential for interpreting the results of future experiments designed to evaluate the time course of 3D sensory and sensorimotor processing and the potential consequences of neuron, session, and/or condition variability in the timing of choice-related activity. Surprisingly, few studies have used reaction time tasks to assay 3D vision. The potential for future studies using reaction time tasks to investigate these important questions are now discussed in subsection “Sensorimotor and choice-motor associations”.

2) It is potentially of strong interest that there is a correspondence between tilt and choice selectivity as shown in 5G. A key question is whether this correspondence arose because of training, or because this could be a naturally occurring correspondence. It is possible that eye movements are naturally drawn to the closest part of a tilted 3D object. It would be very interesting to test in naïve animals whether this is indeed the case (perhaps during passive presentation of tilted objects and free viewing). If there is no such effect in naïve animals, but it is observed in the trained animals, this would support the idea that the correspondence arose because of training. Was the tuning assessed for each neuron during passive viewing outside the context of the task? If so, this might be a way to get at the question by comparing task vs passive viewing.

The reviewers raise a great question regarding whether the correspondence between sensory and choice selectivity is naturally occurring or due to training. We buried too deeply (mostly in the Discussion of the original manuscript) that the study was designed to distinguish between these possibilities through a comparison with our recent CIP work (Elmore et al., 2019). Specifically, the current task was to report the near side of the plane whereas the previous task was to report the far side. By having monkeys report opposite features of the same surface orientation, we were able to dissociate the stimulus from the choice report/saccade direction. For example, for a bottom-near plane, the correct response was the lower choice target in this study but the upper choice target in the previous study. This difference allowed us to evaluate if the association reflected the task training. Importantly, the choice and surface orientation preferences aligned in both cases according to the training. For example, consider a neuron that preferred bottom-near planes. If trained to report the near side, the preferred choice report direction was typically down. In contrast, if trained to report the far side, the preferred choice report direction was typically up. Thus, the correspondence was flexible and experience dependent, and did not simply reflect a tendency for the eyes to be drawn to the closest part of the surface.

To give this finding a more prevalent place in the manuscript, we now present choice-conditioned tuning curves (following several recent papers including Elmore et al., 2019) which show that the same choice report direction was associated with opposite effects on neuronal activity depending on the training. These findings are presented in subsection “Choice-related activity was parametrically tuned and aligned with the tilt preferences”, in new Figure 5—figure supplement 2, and further discussed in subsection “Associations between 3D selectivity and choice-related activity”. We also note that a full comparison of responses measured during the 8AFC task versus passive viewing would be impractical due to the large number of 3D pose conditions (N = 132) and would likely yield ambiguous results due to the training (e.g., it would be difficult to rule out that choice-related processes were not automatically engaged even during passive viewing).

3) The reviewers agreed with one another that the results are described throughout the manuscript in a way which implies a causal relationship between CIP activity/encoding and choice. But given the correlational approaches used here, this message should be revised and clarified in order to avoid confusion and to better bring the discussion of the paper in line with the observed results and experimental design.

We agree and note that it was not our intention to imply a causal relationship. Thank you for bringing this to our attention. We made revisions throughout the text to assiduously avoid any language that might lead to this interpretation. For example, we no longer refer to any process or function as potentially mediating any other. Instead, we refer to discovered *associations* between various sensory, choice-, and motor-related properties. We correspondingly changed the manuscript title to “Functional links between sensory representations, choice activity, and sensorimotor associations in parietal cortex”.

4) All reviewers were in agreement that the writing of the manuscript needs to be improved substantially in order to help readers follow the logic of the analyses, experimental design, and results.

Thank you for bringing this to our attention. As suggested, we made substantial revisions throughout the manuscript to help readers more easily follow the logic of the analyses, experimental design, and results. For example, we expanded the opening statements in each section of the Results to clarify the question(s) driving the upcoming analyses. We also made organizational changes to improve the overall flow. We believe that these revisions substantially improved the manuscript.

5) Analysis related to Figure 4: authors should clarify which period of the task they are describing here, is it SO or SPC?

We now clarify in both the text (subsection “Neuronal correlates of 3D tilt sensitivity”) and figure legend that neuronal data analyzed in Figure 4 is from the SD window.

They separate sensory (SO) and choice response (SPC) periods. The distinction between SO and SPC seems pretty arbitrary.

As discussed in Essential Revision 1, we revised the text to clarify the rationale behind this distinction. Here we go into greater detail about the analyses we incorporated to confirm that the SD and SPC windows defined reliable, functionally distinct response epochs. Some of these points are discussed further in Essential Revisions 1 and 6 relating to the SD/SPC windows, state-space analysis, and dissociating choice- from saccade-related activities.

1) We confirmed at the individual neuron level that significant choice-related activity was rare in the SD window (22/437 neurons, 5%, and thus consistent with the expected rate of false positives) and not linked to a neuron being classified as having choice-related activity since only 13 had choice-related activity in the SPC window. Excluding these neurons did not affect the mean Tolerances of neurons with/without choice-related activity in either time window. These results are presented in subsection “Choice-related activity was parametrically tuned and aligned with the tilt preferences”.

2) We confirmed that choice tuning curves calculated over the full SPC window (from the onset of choice-related activity to the end of the stimulus presentation) were highly similar to those calculated over a short SPC window matching the duration of the SD window. This analysis indicates that the full SPC window was sufficient to characterize choice tuning and its relationship to surface tilt selectivity. These results are presented in subsection “Choice-related activity was parametrically tuned and aligned with the tilt preferences” and in new Figure 5—figure supplement 1A-D.

3) We performed a state-space analysis (demixed principal component analysis) which provided a second estimate of the time course of choice-related activity. Based on this analysis, choice-related activity was not detectable until 232 ms (note that we used the first, more stringent, estimate of 202 ms for our analyses). Importantly, this corroborates that choice-related activity was not detectable in the SD window. These results are presented in subsection “Choice-related activity was parametrically tuned and aligned with the tilt preferences” and in new Figure 5—figure supplement 1E,F.

4) We performed a parametric analysis of the shape of the 3D orientation tuning curves (for details, please see Essential Revision 6). These results showed that the orientation tuning curves of neurons with/without choice-related activity were similar during the SD window. Changes in the shape of the tuning curves across the SD and SPC windows were greater for neurons with than without choice-related activity, supporting the distinction between the two neuronal subpopulations and the two time windows. These findings and their interpretation are presented in subsection “Choice-related activity was associated with more robust 3D selectivity” and in new Figure 7—figure supplement 1.

5) We found that increases in the robustness of 3D selectivity (i.e., increased tolerance of 3D orientation selectivity to distance) across the time windows was associated with choice- but not saccade-related activity. These findings are presented in subsection “Distinguishing functional links between the robustness of 3D selectivity and choice versus saccade-related activity”, in new Figure 10, and further discussed in subsection “Sensorimotor and choice-motor associations”.

The only neuronal activity they show up to Figure 5 are heat map. It would be helpful to see the actual time course of the neuronal response for example neurons.

Thank you for the suggestion. As we discuss in Essential Revision 6, we now show response time courses for preferred and non-preferred stimuli in new Figure 3—figure supplement 1A.

6) Although they show that 5 example neurons, they do not provide clear population analyzes of 3D orientation selectivity. At least, they should show the distribution of tolerance index for all the neurons.

We agree that it would be beneficial to include more details regarding 3D orientation selectivity across the population. In particular, how that selectivity differed for neurons with/without choice-related activity and how it differed across the time windows. We therefore incorporated three new analyses. Note that all of the population analyses included in the original manuscript remain, including: (1) distributions of 3D orientation preferences at each distance in Figure 3F, (2) histograms showing the proportion of neurons with orientation tuning as a function of distance and the number of distances in Figure 6F,G, and (3) Tolerance distributions comparing neurons with/without choice-related activity in Figure 7A-C.

1) As suggested, we added Tolerance distributions for all neurons in each time window in new Figure 3—figure supplement 1B,C. These are the same data as in Figure 7A-C but presented agnostically to whether or not the neurons had choice-related activity. Pointers to these plots are found in subsection “The representation of 3D surface pose by CIP neurons” and subsection “Choice-related activity was associated with more robust 3D selectivity”.

2) We computed a tilt discrimination index (TDI) to assess how well preferred and nonpreferred tilts could be discriminated using single neuron responses. The TDI values were compared to the behavioral tilt sensitivities as a function of slant and distance. This analysis shows that neuronal tilt discriminability and behavioral tilt sensitivity closely mirror each other. The findings are presented in subsection “Neuronal correlates of 3D tilt sensitivity” and in Figure 4A,B.

3) We conducted a parametric analysis of the orientation tuning curve shapes. This analysis tested for differences in the orientation tuning of neurons with/without choice-related activity and differences in tuning across the SD/SPC windows. In brief, bandwidths did not differ between the two neuronal subpopulations or across the time windows. The tuning curves of both subpopulations were more isotropic (less elongated) in the SPC than the SD window, indicating that tilt and slant tuning widths became more similar later in the response. Neurons with choice-related activity showed greater changes in tuning isotropy than neurons without choice-related activity. These findings and their interpretation are presented in subsection “Choice-related activity was associated with more robust 3D selectivity” and in new Figure 7—figure supplement 1.

Given the response shown in Figure 5A, I wonder how neurons could be tilt selective during the SO period. I'd like to see the actual response of these neurons.

We believe that there was some confusion here, and therefore expanded the text describing the quantification of choice-related activity in subsection “Choice-related activity was parametrically tuned and aligned with the tilt preferences”. Specifically, the assessment of choice-related activity was performed using frontoparallel planes only because tilt is undefined at that orientation. Because Figure 5A shows responses to frontoparallel planes, there is no tilt selectivity in either the SD or SPC window. Differences between the curves during the SPC window reflect choice-related activity. To further clarify this point, we now show response time courses for preferred and non-preferred stimuli (across all tilts, slants, and distances) in new Figure 3—figure supplement 1A. We also report that the population responses to preferred and non-preferred stimuli became significantly different 58 ms after stimulus onset. This is only 6 ms after the median visual response latency, indicating that 3D pose selectivity started early in the responses (subsection “The representation of 3D surface pose by CIP neurons”).

7) There is confusion regarding the discussion of vergence, raised by reviewer 3, which needs to be addressed.

Thank you for drawing this confusion to our attention. In the revised manuscript we clarify that this test evaluated if the neuronal responses were significantly affected by small vergence eye movements that did not violate the vergence window. As suggested, we moved the analysis so that it would not distract from the 3D pose tuning results. It is now located at the end of the Materials and methods.

Also, the correlation of tilt selectivity between SO and SPC shown in Figure 7 needs to be clarified and explained in more detail.

We completely agree. To clarify how we quantified differences in tilt preferences across distance in the SD and SPC windows, we added a schematic which illustrates each step of the analysis (new Figure 7D). We also expanded the corresponding text in subsection “Choice-related activity was associated with more robust 3D selectivity”. Pointers to the Materials and methods direct readers to the mathematical details.

[Editors' note: further revisions were suggested prior to acceptance, as described below.]

The manuscript has been improved but there are some remaining issues that need to be addressed before acceptance, as outlined below:1) The new analysis with previous data required additional clarification and explanation, and perhaps additional analysis. The reviewers agreed that there is concern about whether the observed results can be explained by saccade preparation. Specifically one reviewer states that "I don't understand how 3D selectivity and choice can be aligned as shown in Figure 5 considering that choice changes depending on the training (Figure 5—figure supplement 1). If a neuron is tuned to right forward, and monkey is trained to choose forward, the neuron will fire strongly during SPC when stim is right forward and therefore when monkey is planning a saccade to the right. When trained to report away, the same neuron should, according to Figure 5—figure supplement 1, responds strongly during SPC when planning a saccade to the left. How is that possible given that 3D selectivity is getting stronger during SPC and that 3D selectivity and saccade spatial selective strongly overlap (Figure 8)? I understand that they are not analyzing the same neurons, but then, does it only mean that choice selectivity in CIP reflects saccade preparation? I'm genuinely afraid I am missing something important here."

The four conditions outlined in Reviewer 3’s second comment below were helpful for understanding where the confusion arose regarding the alignment of tilt and choice report/saccade signals. Please see our response to that comment for further clarification and pointers to text edits. Briefly, choice report/saccade direction and planar surface tilt are 360° periodic variables. As matters of convention, 0° corresponds to a rightward choice report/saccade direction as well as right-side near planes. Given these conventions and a learned association between surface tilt and choice report/saccade directions, we found the following results: (1) if the monkey was trained to report near, then the surface tilt and choice report/saccade direction preferences had the same value, and (2) if the monkey was trained to report far, then the surface tilt and choice report/saccade direction preferences were 180° apart. That is, the alignment (being either “in phase” or “out of phase”) depended on the specific training, as shown in Figure 5—figure supplement 2.

Importantly, several lines of evidence supported that the choice- and saccade-related activities were functionally distinct. First, we found systematic differences in the choice and saccade tuning curve bandwidths of individual neurons. Second, the choice- and saccade-related activities were dissociable across the population. Specifically, many neurons exhibited significant saccade direction tuning only or choice tuning only (not both). We incorporated additional text emphasizing these findings in the Results and Discussion. Third, we found that the robustness of 3D selectivity was associated with choice-related activity but not saccade-related activity. For that analysis, we divided neurons into four subpopulations based on whether or not they had choice-related activity (assessed using responses to the ambiguous frontoparallel plane stimuli) and whether or not they had saccade-related activity (assessed using the visually guided saccade task, without the presentation of planar surfaces). Neurons with saccade-related activity but not choice-related activity did not show an increase in the robustness of 3D selectivity across time windows. Thus, a change in 3D selectivity could not be explained by saccade preparation signals alone. In contrast, neurons with choice-related activity showed an increase in the robustness of their 3D selectivity across the time windows. Importantly, the change in 3D selectivity did not depend on whether or not the neurons had saccade-related activity. Thus, we found no association between saccade preparation signals and improvements in 3D selectivity. These results further support that choice- and saccade related activities did not simply reflect a common signal. We incorporated additional text to further clarify these points in the Results and Discussion.

Reviewer #1:In reviewing this revised manuscript, the authors have done a thorough job in responding to the previous round of review. They have clarified key points, and in particular largely resolved major concerns about whether the observed association like effects were indeed due to training by including an additional supplementary figure, and by more thoroughly referring to previous results from their group which help address this concern.

We thank the reviewer again for their feedback. The comments helped us substantially improve the manuscript.

Reviewer #2:The authors did an excellent job in revising the manuscript. Specifically, the comparison with the task where the monkeys had to indicate the opposite features of the stimulus is crucial to demonstrate that the association between stimulus orientation and choice direction was indeed learned. I have no further major comments.Reviewer #3:I recognize authors spend a lot of energy trying to address our comments. I am not sure they fully manage to address them though.Here are the main claims (Abstract):• Choice related activity (saccade) is carried by neurons with robust 3D selectivity.• Following onset of choice activity 3D robustness increased• 3D selectivity and saccade selectivity alignedHere are my main concerns:1) Specifically, their analysis consisting of testing for choice related activity in the SD window is not convincing. Our point was that some choice related activity could start before the end of SD window and before the start of the SPC window, not that the activity during the entire SD window was related to choice. In this regard, they failed to address our concerns.

In addition to the ANOVA-based analysis found at line 382, which confirmed that choice-related activity was not detectable during the sensory dominant (SD) window for individual neurons, we verified that choice-related activity was not detectable in the SD window at the population level using: (1) an ANOVA-based analysis and (2) a state-space analysis combined with a choice classification procedure. We still used “dominant” when defining the window to reflect the possibility that a small amount of choice-related activity went undetected, perhaps starting just before the end of the SD window. We further clarified this point in the Results. Importantly, our main analyses focused on changes in 3D selectivity between the two time windows which differed in the amount of choice-related activity present. As such, any choice-related activity in the SD window would only reduce the observed differences. The fact that we found significant changes in 3D selectivity across the two time windows suggests that if any choice-related activity were in the SD window then it did not affect our conclusions.

2) I am still very confused by Figure 5—figure supplement 2 and the claim that 3D selectivity and saccade/choice selectivity are aligned:Let's consider a neuron tuned to top near. We have 4 conditions based on task design (near-CN-or far-CF) and stimulus slant (Top Near-TN-or Bottom near-BN): a) CN/TN, b) CN/BN, c) CF/TN, d) CF/BN.Condition a: choose near with stimulus top near: this neuron will have a strong visual response which will stay sustained since it is aligned to motor plan.Condition b: Choose near with stimulus bottom near: this neuron will have a weak visual response, which will stay weak during motor planning.Condition c: choose far with stimulus top near: there will be a strong visual response (similar to a, neuron's favorite stim) which should decrease during motor planning.Condition d: choose far with stimulus bottom near (saccade to the top): there will be a weak visual response (similar to b, neuron's less favorite stim) which should increase during motor planning.I am confused how this is translated to Figure 5 sp2 and how does that match authors' claim that "Following onset of choice activity 3D robustness increased" and that choice and "3D selectivity and saccade selectivity are aligned". Separating between SC and SPC window does not help at all to understand the neuronal response properties.

Thank you for laying out these four conditions. They helped us to understand where the confusion arose. We made corresponding text edits to clarify the relevant points in the Results and Discussion. As we discuss in our response preceding the reviewer comments, the confusion seems to revolve around the conventions for describing choice report/saccade directions and surface tilt.

We found that the alignment was reversed if the monkey was trained to report the near versus far side of the plane, indicating that the association was experience dependent. In the following description, note that 90° indicates top-near planes as well as upward choice reports/saccades. If the monkey was trained to report near, a preference for top-near planes (tilt = 90°) aligned with a preference for upward choice reports (i.e., saccade to the target at 90° which was used to indicate tilt = 90°). In contrast, if the monkey was trained to report far, a preference for top-near planes (tilt = 90°) aligned with a preference for downward choice reports (i.e., saccade to the target at 270° which was used to indicate tilt = 90°). As such, conditions c and d outlined above are incorrect. For condition c, the stimulus response would be strong as indicated but so would the choice component since downward choices indicated that the bottom was far. For condition d, both the stimulus response and choice component would be weak since they correspond to the non-preferred surface tilt and non-preferred choice report direction.

3) Their attempt to dissociate choice and motor planning is not successful. Stimuli and motor action are still strongly linked whether monkeys have to report near or far side. I don't see how it breaks the link.

In the revised manuscript, we elaborated on findings which support that choice- and saccade-related activities are dissociable. Briefly, the dissociation was supported by three findings: (1) systematic differences in the choice and saccade tuning bandwidths of individual neurons, (2) dissociation of choice- and saccade-related activities across the population (i.e., many neurons were tuned either for choice or saccade direction only, not both), and (3) an increase in the robustness of 3D selectivity across the time windows was only associated with choice-related activity, not saccade-related activity. If choice- and saccade-related activities simply reflected a common signal, saccade-related activity would likely have also been associated with the robustness of 3D selectivity, which was not the case. Please see our response preceding the reviewer comments for further details and pointers to text edits.

A main finding of the study is that there is indeed a sensorimotor association linking the stimuli and motor response. The experiment referenced suggests that the association is experience dependent. We made additional text edits to further clarify that the comparison of results obtained with tasks in which the monkeys reported opposite stimulus features demonstrates that the association between surface tilt and the monkey’s response direction was experience dependent. Please see our responses above related to Figure 5—figure supplement 2 for further details and pointers to text edits.

4) Despite their effort to improve the flow of argument in the article, I still find it very difficult to follow.

We thank the reviewer for the additional feedback, and hope that the additions summarized above have resolved the remaining points of confusion.